# Belief Projection-Based Reinforcement Learning for Environments with Delayed Feedback

**Jangwon Kim**[1]
jangwonkim@postech.ac.kr

**Hangyeol Kim** [2]
hangyeol.kim@koreaaero.com

**Jiwook Kang**[2]
jiwook.kang@koreaaero.com

**Jongchan Baek**[3]
jcbaek@etri.re.kr

**Soohee Han**[1]
soohee.han@postech.ac.kr

## Abstract

We present a novel actor-critic algorithm for an environment with delayed feedback, which addresses the state-space explosion problem of conventional approaches. Conventional approaches use an augmented state constructed from the last observed state and actions executed since visiting the last observed state Using the augmented state space, the correct Markov decision process for delayed environments can be constructed; however, this causes the state space to explode as the number of delayed timesteps increases, leading to slow convergence. Our proposed algorithm, called Belief-Projection-Based $Q$-learning (BPQL), addresses the state-space explosion problem by evaluating the values of the critic for which the input state size is equal to the original state-space size rather than that of the augmented one. We compare BPQL to traditional approaches in continuous control tasks and demonstrate that it significantly outperforms other algorithms in terms of asymptotic performance and sample efficiency. We also show that BPQL solves long-delayed environments, which conventional approaches are unable to do.

## 1 Introduction

Deep reinforcement learning (RL) methods have been successfully applied to diverse domains, such as decision-making tasks and robotic control problems [23, 2]. The deep RL framework has shown its potential by mastering the highly complex board game Go and defeating the professional Go player Lee [30]. It has also demonstrated superhuman performance, even in games where a perfect model is not provided, such as Atari video games [24]. Furthermore, modern deep RL algorithms have achieved significant performance improvements in continuous control domains, such as robotic locomotion [28, 29, 14]. Recently, many attempts have been made to apply RL-based control in not only simulations but also the real-world domain [13, 16, 22, 26]. Remarkable examples include applying the RL method to a quadrupedal robot to learn walking gaits and plan the motion of robotic manipulation in the real world [14, 17].

Despite recent progress in RL methods, adapting RL algorithms to the real world remains challenging for many reasons. Delayed feedback from the environment is a particular challenge. For example, latency may occur when the controller attempts to communicate with an agent if the agent is located far from the controller or if a large quantity of data, such as high-resolution images, must be transmitted. Furthermore, hardware issues can cause sensing or actuator delays. This delayed feedback may hinder achieving the control objectives.

[1]Computational Control Engineering Lab., Pohang University of Science and Technology. [2] Korea Aerospace Industries, Ltd. [3] Electronics and Telecommunications Research Institute.

37th Conference on Neural Information Processing Systems (NeurIPS 2023).

Therefore, controlling signal delay is essential for applying RL algorithms to real-world domains. In this study, we propose a novel approach called belief projection-based $Q$-learning (BPQL) to overcome a constant delayed environment in which feedback is delayed by fixed timesteps. BPQL addresses the state-space explosion problem, which commonly occurs in conventional approaches, and achieves better performance than conventional approaches in terms of asymptotic performance and sample efficiency.

## 2 Related Work

### 2.1 Standard Reinforcement Learning

A Markov decision process (MDP) is defined as a 5-tuple $(\mathcal{X}, \mathcal{A}, R, P, \gamma)$, where $\mathcal{X}$ is the state space, $\mathcal{A}$ is the action space, $R : \mathcal{X} \times \mathcal{A} \mapsto \mathbb{R}$ is the reward function, $P : \mathcal{X} \times \mathcal{A} \times \mathcal{X} \mapsto [0, 1]$ is the transition kernel, and $\gamma \in (0, 1)$ is a discount factor. The policy $\pi(\cdot|s)$ maps the state-to-action distribution. In the standard RL framework, the agent chooses an action to maximize the discounted cumulative rewards by interacting with an environment defined as an MDP.

Let the distribution of the initial state be $\rho_0$. Then, the expected discounted cumulative reward from policy $\pi$ is given as:

$$\eta(\pi) = \mathbb{E}[\sum_{t=0}^{\infty} \gamma^t R(s_t, a_t)], \tag{1}$$

where initial state $s_0 \sim \rho_0, a_t \sim \pi(\cdot|s_t)$, and $s_{t+1} \sim P(\cdot|s_t, a_t)$

From the standard definitions, the value and $Q$-value functions are obtained as:

$$V^\pi(s_t) = \mathbb{E}[\sum_{k=0}^{\infty} \gamma^k R(s_{t+k}, a_{t+k})|s_t] \tag{2}$$

$$Q^\pi(s_t, a_t) = \mathbb{E}[\sum_{k=0}^{\infty} \gamma^k R(s_{t+k}, a_{t+k})|s_t, a_t], \tag{3}$$

where $\forall k \geq 0, a_{t+k} \sim \pi(\cdot|s_{t+k})$ and $s_{t+k+1} \sim P(\cdot|s_{t+k}, a_{t+k})$.

Both value functions satisfy the Bellman equation [3]:

$$V^\pi(s_t) = \mathbb{E}_{a_t \sim \pi(\cdot|s_t)} \left[ R(s_t, a_t) + \gamma \mathbb{E}_{s_{t+1} \sim P(\cdot|s_t, a_t)} \left[ V(s_{t+1}) \right] \right] \tag{4}$$

$$Q^\pi(s_t, a_t) = R(s_t, a_t) + \gamma \mathbb{E}_{s_{t+1} \sim P(\cdot|s_t, a_t)}[V(s_{t+1})]. \tag{5}$$

In the standard RL framework, we assume that the reward and next state only depend on the immediately previous state and action based on the Markov assumption. However, in an environment where the observation is delayed, the next state does not depend on the immediately previous state but on an older state and action history. This delay property forces the environment to be a partially observable MDP, not an MDP.

In this study, we consider the observation-delayed environment but not the action-delayed environment. However, observation and action delays are intrinsically linked [18, 34], which allows us to control the action-delay problem in the same manner as the observation-delay problem.

### 2.2 Constant Observation-Delayed Environment

In a delayed feedback environment, the agent receives time-delayed feedback from the environment, which makes it difficult to choose a timely and correct action at each timestep. In conventional control theory, signal delay is often handled by augmenting the state as a combination of the last observed state and history of actions for the delayed timesteps [20, 25].

This augmented state can also be used in RL frameworks. Considering a constant delayed feedback system, an environment with delayed feedback can be formalized by a constant delayed MDP (CDMDP) [34]. The CDMDP is defined as a 6-tuple $(\mathcal{X}, \mathcal{A}, R, P, \gamma, d)$, where $d$ is the number of timesteps between the execution of an action and receipt of feedback from that action (i.e., delayed timesteps).

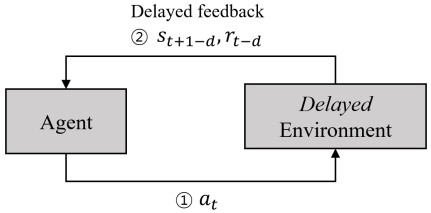

Figure 1: In a constant delayed environment, the agent receives delayed state $s_{t+1-d}$ and reward $r_{t-d}$ after executing action $a_t$. These values are the actual feedback from previous action $a_{t-d}$ and state $s_{t-d}$, where $d$ is the number of *delayed* timesteps.

The CDMDP is known to be reducible to an MDP $(\mathcal{S}, \mathcal{A}, \bar{\mathcal{R}}, \bar{P}, \gamma)$, where $\mathcal{S} = \mathcal{X} \times \mathcal{A}^d$ is an augmented state space, $\bar{\mathcal{R}} : \mathcal{S} \times \mathcal{A} \mapsto \mathbb{R}$ is the reward mapping, and $\bar{P} : \mathcal{S} \times \mathcal{A} \times \mathcal{S} \mapsto [0, 1]$ is the transition kernel [18, 34]. This makes it possible to treat the CDMDP as a regular MDP. From this perspective, some model-free approaches, referred to as *augmented state approaches*, have been proposed [4, 34, 27, 6]. These use the augmented state, which is constructed by concatenating the last observed state and previous actions since visiting the observed state. However, the augmented state space grows exponentially as the number of delayed timesteps $d$ increases. The *curse of dimensionality* makes the augmented state approach tractable for environments with only a relatively short delay.

Model-based approaches have also been proposed for handling delayed environments [34, 15, 11, 8, 1]. In these approaches, the agent predicts the current state by simulating the environment model and selects an action based on the predicted state. Increasing the accuracy of the transition model is key to this approach. However, in a complex, stochastic environment with a long time delay, the rapid growth of model error increases the difficulty of training the agent.

To the best of our knowledge, the closest work to ours is the study [1] that proposes a model-based algorithm expectation maximization $Q$-learning (EMQL). This algorithm, however, can only be used in discrete states and action spaces, whereas our proposed algorithm can be used in continuous state-action spaces without constraints. Additionally, although EMQL can suffer from dynamic model errors, our algorithm avoids this because it is a completely *model-free* approach.

As aforementioned, delayed environments have been mainly handled with two methods: the complete information method (e.g., the augmented state method) and the state estimation methods (e.g., the model-based method). As inherent drawbacks, the first one suffers from the curse of dimensionality and the second requires accurate model-based state estimation. Therefore, the research objective of this work is to resolve these two drawbacks together, i.e., find a model-free approach that is basically free of dimensionality curses. This is why the proposed algorithm was born.

## 3 Dynamics in Augmented State Space

In this section, we analyze the augmented state approach for the constant delayed environment. Figure 1 illustrates the interaction between an agent and a delayed environment. At each time $t$, the agent executes action $a_t$ and receives the delayed state $s_{t+1-d}$ and reward $r_{t-d}$, where $d$ is the number of delayed timesteps. These feedback values are the response from action $a_{t-d}$. This misalignment of state, action, and reward makes it difficult to find an optimal policy.

### 3.1 Notation

We have observed that the CDMDP can be treated as a regular MDP by replacing the state space in the CDMDP with the augmented state space. The augmented state is constructed by concatenating the last observed state and previous actions and is denoted by $\bar{s}$.

More formally, in an environment delayed by $d$ timesteps, the augmented state $\bar{s}_t$ is constructed by concatenating the last observed state and previous $d$ actions:

$$\bar{s}_t = (s_{t-d}, a_{t-d}, a_{t-d+1}, \cdots, a_{t-1}).$$ (6)

The policy that receives the augmented state as input is called the *augmented state-based policy*, denoted as $\bar{\pi}(\cdot|\bar{s})$. $\rho^{\bar{\pi}}(\bar{s})$ is the agent's steady-augmented state distribution for augmented state-based policy $\bar{\pi}$. Furthermore, we refer to the value of the augmented state as the *augmented state-based value*, denoted as $\bar{V}^{\bar{\pi}}(\bar{s}_t)$. Similarly, the $Q$-value of the augmented state is called the *augmented state-based Q-value*, denoted as $\bar{Q}^{\bar{\pi}}(\bar{s}_t, a_t)$.

## 3.2 Augmented State Approach

Our objective is to train the agent in a constant delayed environment. In the augmented state approach, a new augmented state is constructed by augmenting the last observed state and previous actions to treat the CDMDP as a regular MDP.

We can apply a modified Bellman operator called the *delay Bellman operator* $\bar{\mathcal{T}}^{\bar{\pi}}$ to determine the augmented state-based value-function. The operator $\bar{\mathcal{T}}^{\bar{\pi}}$ is given as:

$$\bar{\mathcal{T}}^{\bar{\pi}}\bar{V}(\bar{s}_t) \mapsto \mathbb{E}_{a_t \sim \bar{\pi}(\cdot|\bar{s}_t)}\left[\mathbb{E}_{\mathbb{P}(s_t|\bar{s}_t)}\left[R(s_t, a_t)\right] + \gamma\mathbb{E}_{\bar{s}_{t+1} \sim \bar{P}(\cdot|\bar{s}_t, a_t)}\left[\bar{V}(\bar{s}_{t+1})\right]\right], \forall t > d, \quad (7)$$

where $d$ denotes the number of delayed timesteps in the environment. The expected reward is used in the delay Bellman operator because we assume that all feedback is delayed in a constant delayed environment, including the reward, which implies that the exact reward $R(s_t, a_t)$ cannot be determined at time $t$.

The augmented state-based values can be computed by repeatedly applying the delay Bellman operator. After computing the augmented state-based values, the policy is updated to increase these values for all augmented states in the policy-improvement stage.

Although the augmented state-based values can be computed using the delay Bellman operator, the state-space explosion issue follows; the augmented state space grows exponentially as the number of delayed timesteps $d$ increases because the augmented state space $\mathcal{S}$ in the reduced MDP is a Cartesian product of the original state space and $d$ action space (i.e., $\mathcal{S} = \mathcal{X} \times \mathcal{A}^d$). This indicates that the larger the state space, the larger the set of samples required to compute the value of the entire set of states. This makes the augmented state approach impractical for a long-delayed environment.

## 3.3 Alternative Method to Represent the Augmented State-Based Value

Because calculating the value of the augmented state space by applying the delay Bellman operator suffers from the state-space explosion issue, we propose an alternative method for representing the augmented value.

The distribution of the probability of visiting the true current state $s_t$ (which has not yet been observed) depends on the augmented state $\bar{s}_t$. If two successive transition probabilities $\mathbb{P}(s_t|\bar{s}_t^1)$ and $\mathbb{P}(s_t|\bar{s}_t^2)$ are similar, we can assume that the augmented states $\bar{s}_t^1$ and $\bar{s}_t^2$ have similar representative meanings. Based on this assumption, we define the *belief projection* of the augmented values as follows.

**Definition 3.1.** For a policy $\bar{\pi}$, state space $\mathcal{X} = \{s_1, s_2, \cdots, s_i\}$ and augmented state space $\mathcal{S} = \{\bar{s}_1, \bar{s}_2, \cdots, \bar{s}_j\}$, let $\mathbf{B}$, $\bar{\mathbf{V}}$ and $\mathbf{D}$ be:

$$\mathbf{B} = \begin{pmatrix} \mathbb{P}(s_1|\bar{s}_1) & \mathbb{P}(s_2|\bar{s}_1) & \cdots & \mathbb{P}(s_i|\bar{s}_1) \\ \mathbb{P}(s_1|\bar{s}_2) & \mathbb{P}(s_2|\bar{s}_2) & \cdots & \mathbb{P}(s_i|\bar{s}_2) \\ \vdots & \vdots & \ddots & \vdots \\ \mathbb{P}(s_1|\bar{s}_j) & \mathbb{P}(s_2|\bar{s}_j) & \cdots & \mathbb{P}(s_i|\bar{s}_j) \end{pmatrix}, \bar{\mathbf{V}} = \begin{pmatrix} \bar{V}^{\bar{\pi}}(\bar{s}_1) \\ \vdots \\ \bar{V}^{\bar{\pi}}(\bar{s}_j) \end{pmatrix}, \mathbf{D} = \text{diag}(\mathbf{W}),$$

where $\mathbf{W} = [\rho^{\bar{\pi}}(\bar{s}_1), \rho^{\bar{\pi}}(\bar{s}_2), ..., \rho^{\bar{\pi}}(\bar{s}_j)]$ is steady-augmented state probability vector when the agent follows the policy $\bar{\pi}$. Then, the *projected values* $\mathbf{V}_{\text{proj.}}$ are defined as:

$$\mathbf{V}_{\text{proj.}} = \Pi_{\mathbf{w}}\bar{\mathbf{V}}, \quad (8)$$

where the projection operator $\Pi_{\mathbf{w}}$ is $\mathbf{B}(\mathbf{B}^{\top}\mathbf{D}\mathbf{B})^{-1}\mathbf{B}^{\top}\mathbf{D}$.

We refer to matrix $\mathbf{B}$ as the *belief matrix*. If the matrix $\mathbf{B}^{\top}\mathbf{D}\mathbf{B}$ is not invertible, then the inverse operator can be replaced with the Moore–Penrose pseudoinverse operator. We refer to the outcome

from the projection operator $\Pi_{\mathbf{W}}$ as the *belief projection*. The belief projection operator $\Pi_{\mathbf{W}}$ projects a vector in $\mathbb{R}^j$ onto the column space of the belief matrix $\mathbf{B}$ with respect to the weighted Euclidean norm $||\cdot||_{\mathbf{W}}$.

Because the belief projection lies in the column space of the belief matrix, we can use the vector $\mathbf{V}_\beta$ to decompose the projected values as follows:

$$\mathbf{V}_{\text{proj.}} = \mathbf{B}\mathbf{V}_\beta = \mathbf{B} \begin{pmatrix} V_\beta^{\bar{\pi}}(s_1) \\ \vdots \\ V_\beta^{\bar{\pi}}(s_i) \end{pmatrix}. \tag{9}$$

We refer to the elements of $\mathbf{V}_\beta$ as the *beta values*. The vector $\mathbf{V}_\beta$ can also be defined as:

$$\mathbf{V}_\beta = \arg\min_{\mathbf{V}\in\mathbb{R}^i} ||\mathbf{B}\mathbf{V} - \bar{\mathbf{V}}||_{2,\mathbf{W}}. \tag{10}$$

From Equation (9), the augmented state-based value can be decomposed into the sum of the expectation of the beta values over the original states and the residual:

$$\bar{V}^{\bar{\pi}}(\bar{s}_t) = \mathbb{E}_{\mathbb{P}(s_t|\bar{s}_t)} \left[ V_\beta^{\bar{\pi}}(s_t) \right] + \Delta_{\text{residual}}^{\bar{\pi}}(\bar{s}_t), \tag{11}$$

where $t > d$. Similarly, we can extend Equation (9) to the belief projection for the augmented $Q$-values (projected $Q$-values); the *beta Q-values* for a given action $a$ satisfy the following:

$$\bar{Q}^{\bar{\pi}}(\bar{s}_t, a) = \mathbb{E}_{\mathbb{P}(s_t|\bar{s}_t)} \left[ Q_\beta^{\bar{\pi}}(s_t, a) \right] + \delta_{\text{residual}}^{\bar{\pi}}(\bar{s}_t, a) \tag{12}$$

### 3.4 Linear Function Approximation

The augmented state approach is impractical for a long-delayed environment because its state-space size grows exponentially as the number of delayed timesteps increases. In a long-delayed environment, this indicates that directly calculating the augmented state-based $Q$-values by applying the delay Bellman operator requires a significantly large set of samples.

To avoid this issue, instead of calculating the augmented state-based values $\bar{\mathbf{V}}$ directly, we estimate the belief projection $\Pi_{\mathbf{W}}\bar{\mathbf{V}}$. Let $\bar{T}^{\bar{\pi}}$ be a matrix form of the delay Bellman operator $\bar{\mathcal{T}}^{\bar{\pi}}$ i.e., $\bar{T}^{\bar{\pi}}\bar{\mathbf{V}} := \bar{\mathbf{R}} + \gamma\bar{\mathbf{P}}\bar{\mathbf{V}}$, where $\bar{\mathbf{R}} = [\mathbb{E}_{\mathbb{P}(s_t|\bar{s}_1),a\sim\pi(\cdot|\bar{s}_1)}[R(s,a)], \cdots, \mathbb{E}_{\mathbb{P}(s_t|\bar{s}_j),a\sim\pi(\cdot|\bar{s}_j)}[R(s,a)]]^\top$ is a vector consisting of expected rewards and $\bar{\mathbf{P}}$ is the transition matrix. In this setting, focusing on finding a solution $\mathbf{V}_\beta^* = \arg\min_{\mathbf{V}\in\mathbb{R}^i} ||\mathbf{B}\mathbf{V} - \Pi_{\mathbf{W}}\bar{T}^{\bar{\pi}}(\mathbf{B}\mathbf{V})||_{\mathbf{W}}$ could be achieved very efficiently, especially when $|\mathcal{S}| \gg |\mathcal{X}|$.

Note that the values are approximated as linear combinations of the belief matrix and the beta-values. In other words, the belief projection can be seen as a linear function approximator where the feature vector for an augmented state $\bar{s}_k \in \mathcal{S}$ is $[\mathbb{P}(s_1|\bar{s}_k), \mathbb{P}(s_2|\bar{s}_k), \cdots, \mathbb{P}(s_i|\bar{s}_k)]$ and the corresponding parameters are the beta values $[V_\beta^{\bar{\pi}}(s_1), V_\beta^{\bar{\pi}}(s_2), \cdots, V_\beta^{\bar{\pi}}(s_i)]^\top$. In that sense, we can guarantee the contraction of the combined operator $\Pi_{\mathbf{W}}\bar{T}^{\bar{\pi}}$ by using the well-known contraction property of linear function approximation [32, 5].

**Proposition 3.2.** *Let the projection operator onto the column-space of the belief matrix $\mathbf{B}$ with respect to the weighted Euclidean norm $||\cdot||_{\mathbf{W}}$ be $\Pi_{\mathbf{W}}$, where $\mathbf{W} = [\rho^{\bar{\pi}}(\bar{s}_1), \rho^{\bar{\pi}}(\bar{s}_2), ..., \rho^{\bar{\pi}}(\bar{s}_j)]$ is a steady-augmented state probability vector, and the Markov chain be irreducible i,e, $\rho^{\bar{\pi}}(\bar{s}_k) > 0$ for all $k \in \{1, 2, ..., j\}$. Then the combined operator $\Pi_{\mathbf{W}}\bar{T}^{\bar{\pi}}$ is $\gamma$-contraction with respect to $||\cdot||_{\mathbf{W}}$.*

*Proof.* See appendix B. $\qquad\square$

This contraction property of $\Pi_{\mathbf{W}}\bar{T}^{\bar{\pi}}$ guarantees to find a fixed unique solution $\hat{\mathbf{V}}_\beta^* = [\hat{V}_\beta^{\bar{\pi}}(s_1), \cdots, \hat{V}_\beta^{\bar{\pi}}(s_i)]$ by repeatedly applying the combined operator $\Pi_{\mathbf{W}}\bar{T}^{\bar{\pi}}$. This $\hat{\mathbf{V}}_\beta^*$ can be used as an estimator for the true beta values $\mathbf{V}_\beta$, providing a direction for policy update. For example, a new policy $\bar{\pi}_{\text{new}}$ can be obtained by greedily choosing an action, i.e., $\bar{\pi}_{\text{new}}(\bar{s}_t) = \arg\max_{a_t\in\mathcal{A}} \left( \mathbb{E}_{\mathbb{P}(s_t|\bar{s}_t)}[R(s_t, a_t)] + \mathbb{E}_{\bar{P}(\bar{s}_{t+1}|\bar{s}_t, a_t)} \left[ \mathbb{E}_{\mathbb{P}(s_{t+1}|\bar{s}_{t+1})} \left[ \hat{V}_\beta^{\bar{\pi}_{\text{old}}}(s_{t+1}) \right] \right] \right)$, where the $\bar{\pi}_{\text{old}}$ is the policy before improved.

# 4 Actor-Critic Algorithm for Constant Delayed Environment

In the previous section, we presented an iterative method that computes the value function in the smaller state space created by belief projection, rather than in the larger augmented state space. However, despite the favorable convergence property of value function approximation through belief projection, explicitly calculating the belief matrix can be challenging, especially in cases where the augmented space is large or continuous. To address this challenge, we introduce a practical sampling-based reinforcement learning algorithm based on the theoretical insights from the previous section. In this algorithm, the agent *learns* the beta values without the explicit computation of the belief matrix.

First, we define the *delay soft Bellman operator* $\bar{\mathcal{T}}_{\text{soft}}^{\bar{\pi}}$, which is the soft Bellman operator [14] for the delayed environment setting:

$$\bar{\mathcal{T}}_{\text{soft}}^{\bar{\pi}} \bar{Q}^{\bar{\pi}}(\bar{s}_t, a_t) \mapsto \mathbb{E}_{\mathbb{P}(s_t|\bar{s}_t)}[R(s_t, a_t)] + \gamma \mathbb{E}_{\bar{s}_{t+1} \sim \bar{P}, a_{t+1} \sim \bar{\pi}} \left[ \bar{Q}^{\bar{\pi}}(\bar{s}_{t+1}, a_{t+1}) - \alpha \log \bar{\pi}(a_{t+1}|\bar{s}_{t+1}) \right], \tag{13}$$

where $\alpha$ is the temperature parameter. We can compute the augmented state-based soft $Q$-values by repeatedly applying the delay soft Bellman operator.

In the policy improvement stage, the policy is updated towards the exponential of the evaluated augmented state-based soft $Q$-function. The improved policy $\bar{\pi}_{\text{new}}$ can be obtained as [14]:

$$\bar{\pi}_{\text{new}} = \arg \min_{\bar{\pi}'} D_{\text{KL}} \left( \bar{\pi}'(\cdot|\bar{s}_t) \left\| \frac{\exp(\frac{1}{\alpha} \bar{Q}^{\bar{\pi}_{\text{old}}}(\bar{s}_t, \cdot))}{\bar{Z}^{\bar{\pi}_{\text{old}}}(\bar{s}_t)} \right. \right), \tag{14}$$

where $\bar{Z}^{\bar{\pi}_{\text{old}}}(s_t)$ is a normalizing function. This updated policy $\bar{\pi}_{\text{new}}$ guarantees $\bar{Q}^{\bar{\pi}_{\text{new}}}(\bar{s}_t, a_t) \geq \bar{Q}^{\bar{\pi}_{\text{old}}}(\bar{s}_t, a_t)$ for all $(\bar{s}_t, a_t) \in \mathcal{S} \times \mathcal{A}$.

The beta $Q$-values are approximately computed by minimizing the average of weighted squared residual error i.e, $\delta_{\text{residual}}^{\bar{\pi}}$ in Equation (12) as:

$$J_{Q_\beta} = \mathbb{E}_{\bar{s}_t \sim \rho^{\bar{\pi}}, a_t \sim \bar{\pi}} \left[ \mathbb{E}_{\mathbb{P}(s_t|\bar{s}_t)}[Q_\beta^{\bar{\pi}}(s_t, a_t)] - \bar{Q}^{\bar{\pi}}(\bar{s}_t, a_t) \right]^2. \tag{15}$$

Let the augmented $Q$-value be a soft $Q$-value; then, substituting $\bar{Q}^{\bar{\pi}}(s_t, a_t)$ into (15) with the target value in (13), Equation (15) can be expanded to the following equation:

$$J_{Q_\beta} = \mathbb{E}_{\bar{s}_t \sim \rho(\bar{s}), a_t \sim \bar{\pi}} \left[ \mathbb{E}_{\mathbb{P}(s_t|\bar{s}_t)}[Q_\beta^{\bar{\pi}}(s_t, a_t) - R(s_t, a_t)] - \gamma \mathbb{E}_{\bar{s}_{t+1} \sim \bar{P}(\cdot|\bar{s}_t, a_t), a_{t+1} \sim \bar{\pi}}[\bar{Q}^{\bar{\pi}}(\bar{s}_{t+1}, a_{t+1}) \right.$$
$$\left. - \alpha \log \bar{\pi}(a_{t+1}|\bar{s}_{t+1})] \right]^2. \tag{16}$$

Subsequently, we replace the augmented state-based soft $Q$-value with the expectation of beta $Q$-values:

$$\bar{Q}^{\bar{\pi}}(\bar{s}_{t+1}, a_{t+1}) \approx \mathbb{E}_{\mathbb{P}(s_{t+1}|\bar{s}_{t+1})} \left[ Q_\beta^{\bar{\pi}}(s_{t+1}, a_{t+1}) \right] \tag{17}$$

by fitting the augmented $Q$-value to the belief projection. Then, the objective can be rewritten as:

$$J_{Q_\beta} = \mathbb{E}_{\bar{s}_t \sim \rho(\bar{s}), a_t \sim \bar{\pi}(\bar{s}_t)} \left[ \mathbb{E}_{\mathbb{P}(s_t|\bar{s}_t)} \left[ Q_\beta^{\bar{\pi}}(s_t, a_t) - R(s_t, a_t) \right] \right.$$
$$\left. - \gamma \mathbb{E}_{\bar{s}_{t+1} \sim \bar{P}, a_{t+1} \sim \bar{\pi}} \left[ \mathbb{E}_{\mathbb{P}(s_{t+1}|\bar{s}_{t+1})}[Q_\beta^{\bar{\pi}}(s_{t+1}, a_{t+1})] - \alpha \log \bar{\pi}(a_{t+1}|\bar{s}_{t+1}) \right] \right]^2. \tag{18}$$

Now, our objective has changed to finding the beta $Q$-values, where their expectation best represents the augmented state-based $Q$-values. We can also rewrite Equation (14) using the belief projection

instead of using the augmented $Q$-values:

$$\bar{\pi}_{\text{new}} = \arg\min_{\bar{\pi}'} D_{\text{KL}} \left( \bar{\pi}'(\cdot|\bar{s}_t) \left\| \frac{\exp(\frac{1}{\alpha}\mathbb{E}_{\mathbb{P}(s_t|\bar{s}_t)}[Q_\beta^{\bar{\pi}_{\text{old}}}(s_t,\cdot)])}{\bar{Z}^{\bar{\pi}_{\text{old}}}(\bar{s}_t)} \right. \right) \tag{19}$$

$$= \arg\min_{\bar{\pi}'} \sum_{a_t \in \mathcal{A}} \bar{\pi}'(a_t|\bar{s}_t)\Big( \log\bar{\pi}(a_t|\bar{s}_t) - \frac{1}{\alpha}\mathbb{E}_{\mathbb{P}(s_t|\bar{s}_t)}[Q_\beta^{\bar{\pi}_{\text{old}}}(s_t,a_t)] + \log\bar{Z}^{\bar{\pi}_{\text{old}}}(\bar{s}_t) \Big) \tag{20}$$

$$= \arg\min_{\bar{\pi}'} \sum_{a_t \in \mathcal{A}} \bar{\pi}'(a_t|\bar{s}_t)\Big( \log\bar{\pi}(a_t|\bar{s}_t) - \frac{1}{\alpha}\mathbb{E}_{\mathbb{P}(s_t|\bar{s}_t)}[Q_\beta^{\bar{\pi}_{\text{old}}}(s_t,a_t)] + \mathbb{E}_{\mathbb{P}(s_t|\bar{s}_t)}[\log Z^{\bar{\pi}_{\text{old}}}(s_t)] \Big)$$
$$\tag{21}$$

$$= \arg\min_{\bar{\pi}'} \mathbb{E}_{\mathbb{P}(s_t|\bar{s}_t)}\Big[ \sum_{a_t \in \mathcal{A}} \bar{\pi}'(a_t|\bar{s}_t)\big( \log\bar{\pi}(a_t|\bar{s}_t) - \frac{1}{\alpha}Q_\beta^{\bar{\pi}_{\text{old}}}(s_t,a_t) + \log Z^{\bar{\pi}_{\text{old}}}(s_t) \big) \Big] \tag{22}$$

$$= \arg\min_{\bar{\pi}'} \mathbb{E}_{\mathbb{P}(s_t|\bar{s}_t)}\left[ D_{\text{KL}}\left( \bar{\pi}'(\cdot|\bar{s}_t) \left\| \frac{\exp(\frac{1}{\alpha}Q_\beta^{\bar{\pi}_{\text{old}}}(s_t,\cdot))}{Z^{\bar{\pi}_{\text{old}}}(s_t)} \right. \right) \right], \tag{23}$$

where $Z^{\bar{\pi}_{\text{old}}}(s_t)$ is a normalizing function for the distribution $Q^{\bar{\pi}_{\text{old}}}(s_t,\cdot)$. Equation (21) holds because $\log\bar{Z}^{\bar{\pi}_{\text{old}}}(\bar{s}_t)$ and $\mathbb{E}_{\mathbb{P}(s_t|\bar{s}_t)}[\log\bar{Z}^{\bar{\pi}_{\text{old}}}(s_t)]$ are independent of $\bar{\pi}'$.

Using Equation (23), we can update the policy by minimizing the expectation of the Kullback Leibler-divergence of the policy and exponential of the beta $Q$-value. Notably, we need not evaluate the augmented state-based soft $Q$-value, which causes the state-space explosion problem when the delayed timestep is large. By contrast, we evaluate the beta $Q$-values and improve the policy using these values.

The beta $Q$-value is expected to converge more stably and quicker than the augmented state-based $Q$-value because its input state space is considerably smaller than the augmented space, which also helps to obtain an improved policy with higher quality.

In the continuous state setting, the parameterized beta $Q$-function $Q_{\theta,\beta}^{\bar{\pi}}(s_t,a_t)$ can be approximately computed by minimizing the squared-residual in Equation (18) denoted by $J_{Q_\beta}(\theta)$ with the aid of replay memory $\mathcal{D}$ [24]:

$$J_{Q_\beta}(\theta) = \mathbb{E}_{(\bar{s}_t,s_t,a_t,r_t,\bar{s}_{t+1},s_{t+1})\sim\mathcal{D}}\big[\frac{1}{2}(Q_{\theta,\beta}^{\bar{\pi}}(s_t,a_t) - (r_t + \gamma\mathbb{E}_{a_{t+1}\sim\bar{\pi}}[Q_{\theta,\beta}^{\text{tar}}(s_{t+1},a_{t+1})$$
$$- \alpha\log\bar{\pi}_\phi(a_{t+1}|\bar{s}_{t+1})]))^2], \tag{24}$$

where $r_t = R(s_t,a_t)$, $Q_{\theta,\beta}^{\text{tar}}$ is a target network for $Q_{\theta,\beta}^{\bar{\pi}}$ [24], and $\bar{\pi}_\phi$ is a parameterized policy.

In the policy improvement stage, the policy can be trained by minimizing the objective $J_{\bar{\pi}}(\phi)$, which can be formalized from Equation (23) as:

$$J_{\bar{\pi}}(\phi) = \mathbb{E}_{(\bar{s}_t,s_t)\sim\mathcal{D}}\left[ \mathbb{E}_{a_t\sim\bar{\pi}_\phi}\left[ \alpha\log\bar{\pi}_\phi(a_t|\bar{s}_t) - Q_{\theta,\beta}^{\bar{\pi}}(s_t,a_t) \right] \right]. \tag{25}$$

By iteratively minimizing $J_{Q_\beta}(\theta)$ and $J_{\bar{\pi}}(\phi)$, the actor and critic networks can be trained for a delayed environment. This is the BPQL algorithm. Note that the first and second subscripts of $Q_{\theta,\beta}$ refer to the neural network weights and the beta $Q$-values, respectively. Actually, the beta $Q$-values depend on the neural network weights. Nevertheless, $\theta$ and $\beta$ are written together for clearer notation. Only $\theta$ is the variable set to be optimally taken through learning.

In the BPQL algorithm, expanded transition tuples that include augmented states are stored in the replay memory $\mathcal{D}$. To construct the augmented state, we use a temporary buffer $\mathcal{B}$ in which the original states and action history are stored. The details of BPQL are summarized in Appendix A.

## 5 Experiments

We compared the performance of the BPQL algorithm with the following three baselines[4]:

---

[4]Further details of the implementation of these algorithms are given in Appendix D.

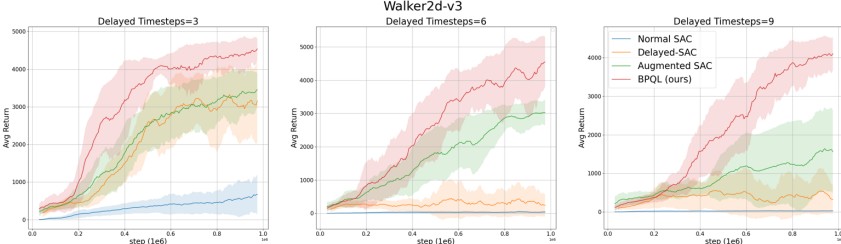

Figure 2: Performance curves of each algorithm for the Walker2d-v3 task. We repeated the test for this task five times with different random seeds. The mean over the results is shown by the solid line, and the shaded area represents the standard deviation. As the delay increases, the proposed algorithm BPQL significantly outperforms other algorithms in terms of asymptotic performance and sample efficiency.

Table 1: Results of MuJoCo benchmark tasks for one million interactions. Each task was evaluated in the delayed environment setting for 3,6, and 9 delayed timesteps $d$. All tests were repeated with five different random seeds. All results are shown with the standard deviation over the five trials denoted by $\pm$.

| Environment | | HalfCheetah-v3 | Walker2d-v3 | Hopper-v3 | Swimmer-v3 | InvertedPendulum-v2 | Reacher-v2 |
|---|---|---|---|---|---|---|---|
| $d$ | Algorithm | | | | | | |
| 3 | Normal SAC | $-276.2_{\pm 93.1}$ | $672.2_{\pm 518.8}$ | $290.9_{\pm 86.7}$ | $32.8_{\pm 3.7}$ | $16.9_{\pm 13.7}$ | $-28.23_{\pm 4.66}$ |
| | Delayed-SAC | $4182.7_{\pm 609.7}$ | $4463.2_{\pm 434.0}$ | $2821.7_{\pm 609.1}$ | $73.9_{\pm 32.6}$ | $916.5_{\pm 142.1}$ | $-3.95_{\pm 0.53}$ |
| | Augmented SAC | $6054.0_{\pm 1045.5}$ | $3453.3_{\pm 462.6}$ | $2732.6_{\pm 858.9}$ | $47.6_{\pm 2.1}$ | $983.6_{\pm 32.6}$ | $-3.95_{\pm 0.51}$ |
| | BPQL (ours) | $\mathbf{8100.1_{\pm 543.4}}$ | $\mathbf{4538.5_{\pm 271.3}}$ | $\mathbf{2922.5_{\pm 671.2}}$ | $\mathbf{88.0_{\pm 36.6}}$ | $\mathbf{998.1_{\pm 3.6}}$ | $\mathbf{-3.80_{\pm 0.51}}$ |
| 6 | Normal SAC | $-288.7_{\pm 50.7}$ | $38.6_{\pm 29.4}$ | $68.3_{\pm 34.1}$ | $32.1_{\pm 6.7}$ | $10.1_{\pm 2.9}$ | $-38.2_{\pm 3.18}$ |
| | Delayed-SAC | $2660.9_{\pm 492.3}$ | $1.0_{\pm 4.0}$ | $1289.2_{\pm 1071.7}$ | $58.2_{\pm 14.6}$ | $929.1_{\pm 141.7}$ | $-4.02_{\pm 0.50}$ |
| | Augmented SAC | $2012.6_{\pm 835.0}$ | $3028.4_{\pm 383.2}$ | $2100.0_{\pm 752.7}$ | $43.4_{\pm 3.8}$ | $\mathbf{1000.0_{\pm 0.0}}$ | $-4.05_{\pm 0.48}$ |
| | BPQL (ours) | $\mathbf{6334.6_{\pm 245.3}}$ | $\mathbf{4551.9_{\pm 759.4}}$ | $\mathbf{3336.0_{\pm 200.3}}$ | $\mathbf{93.0_{\pm 31.8}}$ | $983.4_{\pm 33.1}$ | $\mathbf{-3.81_{\pm 0.51}}$ |
| 9 | Normal SAC | $-294.0_{\pm 46.7}$ | $26.4_{\pm 8.5}$ | $69.0_{\pm 12.5}$ | $26.3_{\pm 3.6}$ | $16.1_{\pm 5.9}$ | $-37.36_{\pm 3.42}$ |
| | Delayed-SAC | $1764.3_{\pm 203.3}$ | $2.6_{\pm 6.4}$ | $513.7_{\pm 642.2}$ | $77.6_{\pm 34.1}$ | $505.8_{\pm 333.2}$ | $-4.01_{\pm 0.51}$ |
| | Augmented SAC | $1297.2_{\pm 265.9}$ | $1562.9_{\pm 1075.9}$ | $1497.8_{\pm 747.7}$ | $38.3_{\pm 4.0}$ | $\mathbf{1000.0_{\pm 0.0}}$ | $-4.39_{\pm 0.54}$ |
| | BPQL (ours) | $\mathbf{5887.5_{\pm 270.5}}$ | $\mathbf{4104.3_{\pm 428.7}}$ | $\mathbf{2993.4_{\pm 566.7}}$ | $\mathbf{93.5_{\pm 34.6}}$ | $985.6_{\pm 28.6}$ | $\mathbf{-3.86_{\pm 0.50}}$ |

- Augmented approach: Augmented SAC
- Model-based approach: Delayed-SAC
- Normal SAC

In the augmented approach, we solve the control problem by adopting an augmented state space, which facilitates treating the CDMDP as a regular MDP. This is a popular and widely used method for an agent learning in a delayed environment [34, 27]. The policy and critic in the augmented state approach are trained by minimizing the objectives $\bar{J}_{\bar{Q}}(\theta)$ and $\bar{J}_{\bar{\pi}}(\phi)$:

$$
\begin{aligned}
\bar{J}_{\bar{Q}}(\theta) = \mathbb{E}_{(\bar{s}_t, a_t, r_t, \bar{s}_{t+1}) \sim \mathcal{D}} \big[ \frac{1}{2} (\bar{Q}_{\theta}^{\bar{\pi}}(\bar{s}_t, a_t) - (r_t + \gamma \mathbb{E}_{a_{t+1} \sim \bar{\pi}}[\bar{Q}_{\theta}^{\text{tar}}(\bar{s}_{t+1}, a_{t+1}) \\
- \alpha \log \bar{\pi}_{\phi}(a_{t+1}|\bar{s}_{t+1})]))^2 \big],
\end{aligned}
\tag{26}
$$

$$
\bar{J}_{\bar{\pi}}(\phi) = \mathbb{E}_{(\bar{s}_t) \sim \mathcal{D}} [\mathbb{E}_{a_t \sim \bar{\pi}_{\phi}}[\alpha \log \bar{\pi}_{\phi}(a_t|\bar{s}_t) - \bar{Q}_{\theta}^{\bar{\pi}}(\bar{s}_t, a_t)].
\tag{27}
$$

We named this augmented-based approach *Augmented SAC*.

In the model-based approach, we use the delayed-Q algorithm [10], which uses a forward model to estimate the timely proper state. In this approach, the agent takes an action based on the predicted current state from the dynamic model using the last observed state and action history, i.e., $(s_{t-d}, a_{t-d}, a_{t-d+1}, \cdots, a_{t-1})$. The model repeats the one-step transition prediction $d$ times to

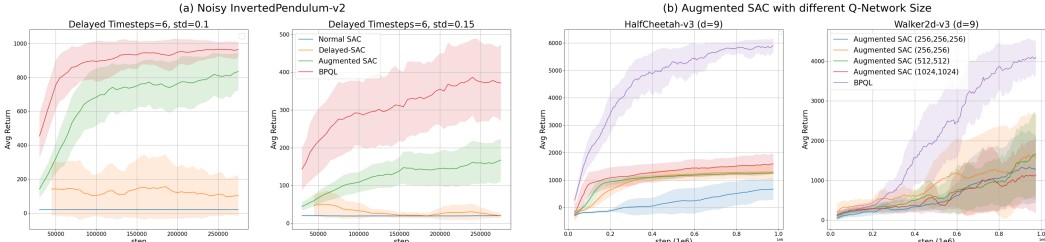

Figure 3: **(a)** Result of tests on noisy version of InvertedPendulum-v2. The noises added to the actuator are drawn from a normal distribution with a mean of zero and standard deviations of 0.1 (*(a)-left*) and 0.15 (*(a)-right*), respectively. The range of action for the actuator is [-3, 3]. **(b)** The performance curves of BPQL and Augmented SAC with different hidden sizes $\in \{(256, 256), (512, 512), (1024, 1024), (256, 256, 256)\}$ of $Q$-network.

predict the current state. In the learning stage, we train the critic and policy using *true* transition tuples, i.e., $(s_t, a_t, r_t, s_{t+1})$. The original study used double deep $Q$-networks (DDQN) [33] as the base learning algorithm. However, because DDQN is only applicable to discrete action spaces, we have used SAC for our base learning algorithm, so named this model-based approach *Delayed-SAC*.

Lastly, in the *Normal SAC* approach, the agent naively uses delayed feedback for training without any modification.

### 5.1 Performance Comparison

We tested the algorithms on several tasks using the MuJoCo benchmark [31] and evaluated their performances in environments with different numbers of delayed timesteps.[5] Figure 2 shows that the augmented and model-based approaches are inappropriate for environments in which the delayed timestep is large, whereas the proposed BPQL algorithm exhibits significantly better performance in a long-delayed environment. Table 1 lists the overall experimental results, confirming that BPQL outperforms the conventional approaches by a wide margin in environments ranging from a short ($d$=3) to a long delay ($d$=9).

### 5.2 Stochastic Environment

We evaluated BPQL and other baselines on the noisy version of the InvertedPendulum-v2 environment, where Gaussian noises were added to the actuator such that the acting becomes stochastic for the same input for the actuator. The added noises to the actuator are sampled from normal distribution with zero mean and standard deviations of 0.1 and 0.15. In this stochastic environment, BQPL has also shown better performance than conventional algorithms, and the difference gap in performance increased as the level of stochasticity grew.

### 5.3 Performance comparison of augmented approach with various capacity of $Q$-network

We conducted additional ablation study to investigate whether the size of $Q$-network was too small to learn and extract important features from the augmented state. We tested the performance of Augmented SAC with different hidden size of the $Q$-network, including (256,256), (512,512), (1024,1024), and (256,256,256). The results of this additional experiment show that the size of the $Q$-network does not significantly affect the performance of the augmented SAC as shown in Figure 3-(b). This confirms that addressing the state space explosion problem is a crucial factor in training the agent in a delayed environment.

## 6 Conclusion

In this study, we proposed a novel model-free algorithm BPQL for a constant delayed environment. BPQL evaluates the beta $Q$-function based on the original state-space size rather than evaluating the

---

[5]See Appendix C for the details of these environments.

augmented $Q$-function, which helps the parameterized $Q$-function learn more stably and converge faster. In regards to the MuJoCo benchmark for continuous control tasks, BPQL achieves significantly better performance than the augmented state approach, which is a popular and widely used algorithm for delayed environments. Our results show that BPQL provides a promising avenue for handling delayed environments, such as real-world robot control environments where sensing and actuator delay exist, or a communication system with a narrow bandwidth.

As BPQL cannot be applied to a random delayed environment, it would be meaningful to extend our work to an environment with randomly delayed feedback. Ensemble learning of the beta $Q$-functions and augmented state-based policies that cover the entire delay range could be one possible approach. Furthermore, real-world applications using the proposed algorithm is an exciting direction for future work.

## Acknowledgement

This work was supported by Institute of Information & communications Technology Planning & Evaluation (IITP) grant funded by the Korea government (MSIT) (No.2019-0-01906, Artificial Intelligence Graduate School Program (POSTECH))

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

# A  Architecture and Pseudo Code of BPQL

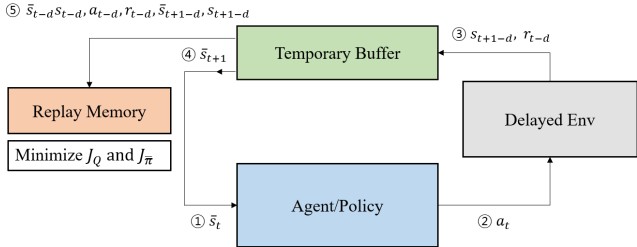

Figure 4: BPQL architecture: The agent selects an action using an augmented state, which is obtained from the temporary buffer, as an input of the policy. At every time $t > 2d$, a transition tuple $\bar{s}_{t-d}, s_{t-d}, a_{t-d}, r_{t-d}, \bar{s}_{t+1-d}, s_{t+1-d}$ is stored in the replay memory, and the policy and beta critic are trained by minimizing $J_{\bar{\pi}}$ and $J_{Q_\beta}$, respectively.

---

**Algorithm 1** Belief-Projection-Based $Q$-learning (BPQL)

---

1: **initialize** a policy $\bar{\pi}_\phi(a|\bar{s})$, beta critics $Q_{\theta,\beta}(s,a)$, target beta critics $Q_{\theta,\beta}^{\text{tar}}(s,a)$, an empty replay memory $\mathcal{D}$, a temporary buffer $\mathcal{B}$, delayed timesteps $d$, learning rate for the beta critic $\lambda_{Q_\beta}$, learning rate for the policy $\lambda_{\bar{\pi}}$, and soft update ratio $\tau$
2: **for** $episode = 1$ to $E$ **do**
3:     **for** $t = 1$ to $H$ **do**
4:         **if** $t < d$ **then**
5:             executes action $a_t$ randomly or 'No-Op' action
6:             put $a_t$ to $\mathcal{B}$
7:         **else if** $t = d$ **then**
8:             choose action $a_d$ randomly or 'No-Op' action
9:             $s_1 \leftarrow$ Env$(a_d)$
10:             put $s_1$ and $a_d$ to $\mathcal{B}$
11:         **else**
12:             get $s_{t-d}, a_{t-d}, \ldots, a_{t-1}$ from $\mathcal{B}$
13:             $\bar{s}_t \leftarrow (s_{t-d}, a_{t-d}, \ldots, a_{t-1})$
14:             $a_t \leftarrow \pi_\phi(\bar{s}_t)$
15:             $s_{t+1-d}, r_{t-d} \leftarrow$ Env$(a_t)$
16:             put $s_{t+1-d}$ and $a_t$ to $\mathcal{B}$
17:             **if** $t > 2d$ **then**
18:                 get $s_{t-2d}, s_{t-2d+1}, s_{t-d}, a_{t-2d}, \ldots, a_{t-d}$ from $\mathcal{B}$
19:                 $\bar{s}_{t-d} \leftarrow (s_{t-2d}, a_{t-2d}, \ldots, a_{t-d-1})$
20:                 $\bar{s}_{t-d+1} \leftarrow (s_{t-2d+1}, a_{t-2d+1}, \ldots, a_{t-d})$
21:                 store $(\bar{s}_{t-d}, s_{t-d}, a_{t-d}, r_{t-d}, \bar{s}_{t+1-d}, s_{t+1-d})$ in $\mathcal{D}$
22:                 pop $s_{t-2d}, a_{t-2d}$ from $\mathcal{B}$
23:             **end if**
24:         **end if**
25:     **end for**
26:     **for** each gradient step **do**
27:         $\omega \leftarrow \omega - \lambda_{Q_\beta} \nabla_\theta J_{Q_\beta}(\omega)$
28:         $\phi \leftarrow \phi - \lambda_{\bar{\pi}} \nabla_\phi J_{\bar{\pi}}(\phi)$
29:         $\bar{\omega} \leftarrow \tau\omega + (1-\tau)\bar{\omega}$
30:     **end for**
31: **end for**

---

# B Proof of Proposition 3.2

**Proposition 3.2** *Let the projection operator on the column-space of the belief matrix $\boldsymbol{B}$ with respect to the weighted Euclidean norm $||\cdot||_{\boldsymbol{W}}$ be $\Pi_{\boldsymbol{W}}$, where $\boldsymbol{W} = [\rho^{\bar{\pi}}(\bar{s}_1), \rho^{\bar{\pi}}(\bar{s}_2), ..., \rho^{\bar{\pi}}(\bar{s}_j)]$ is a steady-augmented state probability vector, and the Markov chain be irreducible i,e, $\rho^{\bar{\pi}}(\bar{s}_k) > 0$ for all $k \in \{1, 2, ..., j\}$. Then the combined operator $\Pi_{\boldsymbol{W}}\bar{T}^{\bar{\pi}}$ is $\gamma$-contraction with respect to $||\cdot||_{\boldsymbol{W}}$.*

*Proof.* Let $\mathbf{V}^1_{\text{proj.}}$ and $\mathbf{V}^2_{\text{proj.}}$ be an arbitrary vector in $\mathbb{R}^j$, $\bar{P}^{\bar{\pi}}(\bar{s}'|\bar{s})$ be a transition probability of $\bar{s} \to \bar{s}'$ when the agent follow the policy $\bar{\pi}$, and $V^{1,\text{proj.}}_k, V^{2,\text{proj.}}_k$ be the k-th element of the $\mathbf{V}^1_{\text{proj.}}$ and $\mathbf{V}^2_{\text{proj.}}$ respectively.

$$||\Pi_{\mathbf{W}}\bar{T}^{\bar{\pi}}\mathbf{V}^1_{\text{proj.}} - \Pi_{\mathbf{W}}\bar{T}^{\bar{\pi}}\mathbf{V}^2_{\text{proj.}}||^2_{\mathbf{W}} \tag{28}$$

$$= ||\Pi_{\mathbf{W}}(\bar{T}^{\bar{\pi}}\mathbf{V}^1_{\text{proj.}} - \bar{T}^{\bar{\pi}}\mathbf{V}^2_{\text{proj.}})||^2_{\mathbf{W}} \tag{29}$$

$$\leq ||\Pi_{\mathbf{W}}(\bar{T}^{\bar{\pi}}\mathbf{V}^1_{\text{proj.}} - \bar{T}^{\bar{\pi}}\mathbf{V}^2_{\text{proj.}})||^2_{\mathbf{W}} + ||(\mathbf{I} - \Pi_{\mathbf{W}})(\bar{T}^{\bar{\pi}}\mathbf{V}^1_{\text{proj.}} - \bar{T}^{\bar{\pi}}\mathbf{V}^2_{\text{proj.}})||^2_{\mathbf{W}} \tag{30}$$

$$= ||\bar{T}^{\bar{\pi}}\mathbf{V}^1_{\text{proj.}} - \bar{T}^{\bar{\pi}}\mathbf{V}^2_{\text{proj.}}||^2_{\mathbf{W}} \tag{31}$$

$$= ||\bar{\mathbf{R}} + \gamma\bar{\mathbf{P}}\mathbf{V}^1_{\text{proj.}} - (\bar{\mathbf{R}} + \gamma\bar{\mathbf{P}}\mathbf{V}^2_{\text{proj.}})||^2_{\mathbf{W}} \tag{32}$$

$$= \gamma\Sigma^j_{k=1}\rho^{\bar{\pi}}(\bar{s}_k)\left(\Sigma^j_{l=1}\bar{P}^{\bar{\pi}}(\bar{s}_l|\bar{s}_k)\left(V^{1,\text{proj.}}_l - V^{2,\text{proj.}}_l\right)\right)^2 \tag{33}$$

$$\leq \gamma\Sigma^j_{k=1}\rho^{\bar{\pi}}(\bar{s}_k)\Sigma^j_{l=1}\bar{P}^{\bar{\pi}}(\bar{s}_l|\bar{s}_k)\left(V^{1,\text{proj.}}_l - V^{2,\text{proj.}}_l\right)^2 \tag{34}$$

$$= \gamma\Sigma^j_{l=1}\Sigma^j_{k=1}\rho^{\bar{\pi}}(\bar{s}_k)\bar{P}^{\bar{\pi}}(\bar{s}_l|\bar{s}_k)\left(V^{1,\text{proj.}}_l - V^{2,\text{proj.}}_l\right)^2 \tag{35}$$

$$= \gamma\Sigma^j_{l=1}\rho^{\bar{\pi}}(\bar{s}_l)\left(V^{1,\text{proj.}}_l - V^{2,\text{proj.}}_l\right)^2 \tag{36}$$

$$\because \Sigma^j_{k=1}\rho^{\bar{\pi}}(\bar{s}_k)\bar{P}^{\bar{\pi}}(\bar{s}_l|\bar{s}_k) = \rho^{\bar{\pi}}(\bar{s}_l) \text{ by the definition of } \rho^{\bar{\pi}}$$

$$= \gamma||\mathbf{V}^1_{\text{proj.}} - \mathbf{V}^2_{\text{proj.}}||^2_{\mathbf{W}}, \tag{37}$$

where Equality (31) holds by the Pythagorean Theorem, and Inequality (34) follows from the Jensen's inequality. $\qquad\square$

## C  Environment Details

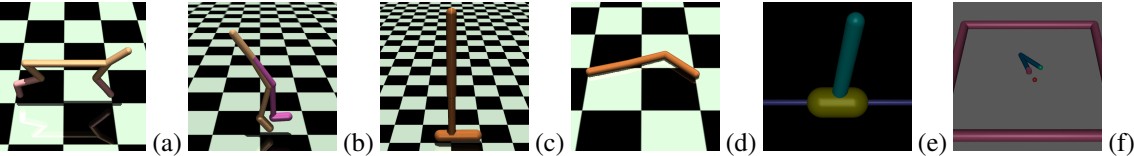

Figure 5: MuJoCo continuous control environments in the experiment: (a) HalfCheetah-v3, (b) Walker2d-v3, (c) Hopper-v3, (d) Swimmer-v3, (e) InvertedPendulum-v2, and (f) Reacher-v2.

Table 2: Details of the MuJoCo environment

|  | Action dimension | State dimension | Timestep (s) |
|---|---|---|---|
| HalfCheetah-v3 | 6 | 17 | 0.05 |
| Walker2d-v3 | 6 | 17 | 0.008 |
| Hopper-v3 (s) | 3 | 11 | 0.008 |
| Swimmer-v3 | 2 | 6 | 0.04 |
| InvertedPendulum-v2 | 1 | 4 | 0.04 |
| Reacher-v2 | 2 | 11 | 0.02 |

## D  Implementation Details

In this section, we provide the implementation details of the algorithms used in this study. All methods presented in the experiment used an SAC as their base learning algorithm with the following characteristics:

- Stochastic Gaussian policy approaches.

- Automated entropy adjustment [14].

- Clipped-double $Q$-learning, which was introduced in the TD3 algorithm to prevent overestimating the $Q$-value [12].

- Soft target update, which changes the target values slowly and improves the learning stability[21].

- Adam optimizer, a variant of the stochastic gradient descent method [19].

The details of the hyperparameters are presented in Table 3.

Table 3: Hyperparameters for BPQL and the baselines.

| Hyperparameters | **Values** |
|---|---|
| Critic network | 256, 256 |
| Policy network | 256, 256 |
| Discount factor | 0.99 |
| Replay memory size | 1 M |
| Minibatch size | 256 |
| Learning rate | 0.0003 |
| Target entropy | -dim$|\mathcal{A}|$ |
| Target smoothing coefficient | 0.995 |
| Optimizer | Adam |

# E   Plots of Performance Comparison

In this section, we present several plots of the performance curves of BPQL and other baselines in the environments with delayed timesteps of 3, 6 and 9.

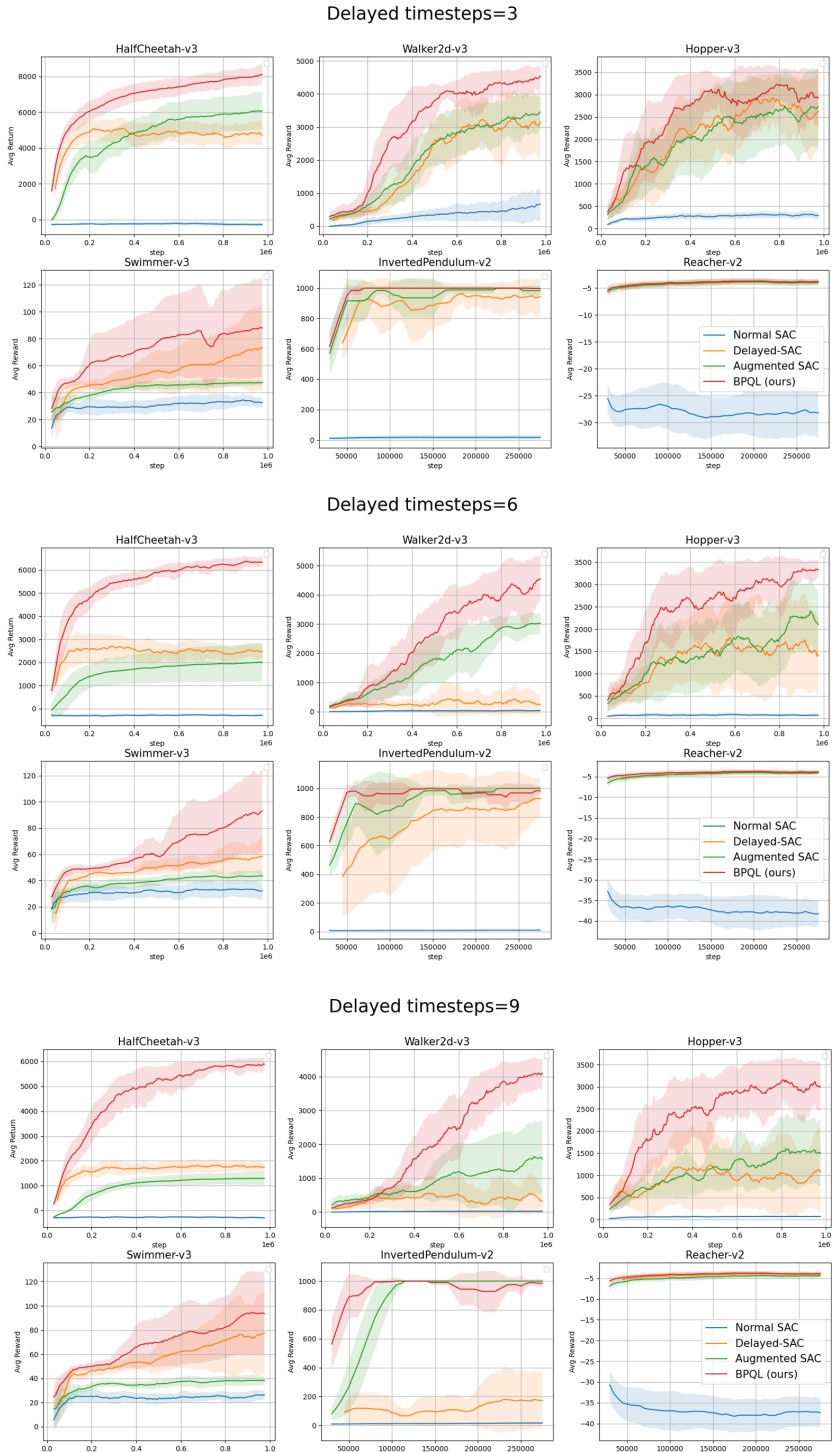

Figure 6: Performance curves for each algorithm in continuous tasks with three, six and nine delayed timesteps.

# F Additional Experiments

## F.1 Extremely Long Delayed Environment

We conducted additional experiments to determine how well BPQL could solve the control problem even in very long delayed environments. Figure 7 shows that in BPQL, the policy is improved through interaction with the environment in a very long delayed environment, but the conventional methods find it difficult improve their policy.

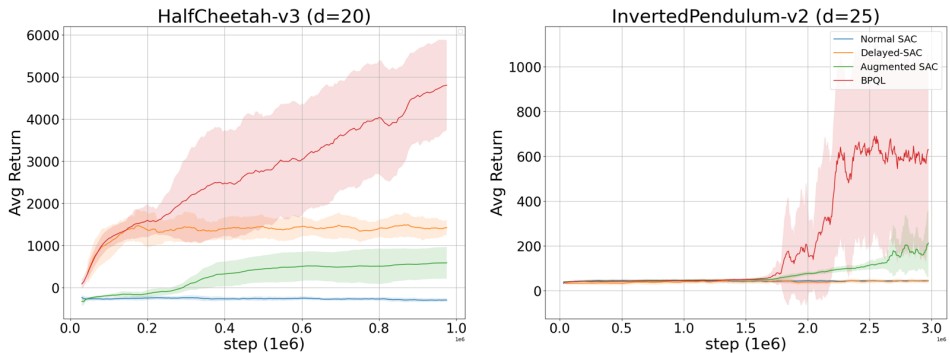

Figure 7: HalfCheetah-v3 environment with 20 delayed timesteps (*left*) and InvertedPendulum-v2 environment with 25 delayed timesteps (*right*). Each timestep is 0.05 s and 0.04 s for HalfCheetah-v3 and InvertedPendulum-v2 environments, respectively. In the InvertedPendulum-v2 environment, to prevent early failure, the agent uses a pretrained policy to determine the actions of the first 25 (=number of delayed timesteps) timesteps. We repeated the test for the tasks five times with different random seeds.

## F.2 Action delay

We also evaluated BPQL and other baselines in *action delayed* environments. Figure 8 illustrates the interaction between an agent and action delayed environment.

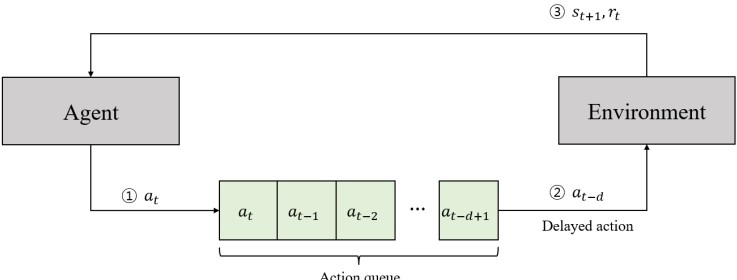

Figure 8: In an action delayed environment, the environment takes the delayed action $a_{t-d}$ instead of the current action $a_t$, where $d$ is the number of delayed timesteps.

### F.2.1 Augmented State in an Action Delayed Environment

In an action delayed environment, the augmented state at time $t + d$ is defined as:

$$\bar{s}_{t+d} = (s_t, a_{t-d}, a_{t-d+1}, \cdots, a_{t-1}),\tag{38}$$

where the $d$ is the number of delayed timesteps for an action. The CDMDP for an action delayed environment is defined as a tuple $(\mathcal{X}, \mathcal{A}, R, P, \gamma, d_a)$, where $\mathcal{X}$ is the original state space, $\mathcal{A}$ is the action space, $R : \mathcal{X} \times \mathcal{A} \mapsto \mathbb{R}$ is the reward function, $P : \mathcal{X} \times \mathcal{A} \times \mathcal{X} \mapsto [0, 1]$ is the transition kernel, $\gamma \in (0, 1)$ is a discount factor, and $d_a$ is the number of delayed timesteps for an action.

This CDMDP can be reduced to a MDP $(\mathcal{S}, \mathcal{A}, \bar{\mathcal{R}}, \bar{P}, \gamma)$, where $\mathcal{S} = \mathcal{X} \times \mathcal{A}^{d_a}$ is an augmented state space, $\bar{\mathcal{R}} : \mathcal{S} \times \mathcal{A} \mapsto \mathbb{R}$ is the reward mapping, and $\bar{P} : \mathcal{S} \times \mathcal{A} \times \mathcal{S} \mapsto [0, 1]$ is the transition kernel as in the case of observation delayed environments [18]. BPQL for action delayed environments uses the augmented state defined in Equation (38) instead of the one defined in Equation (6).

### F.2.2 Results

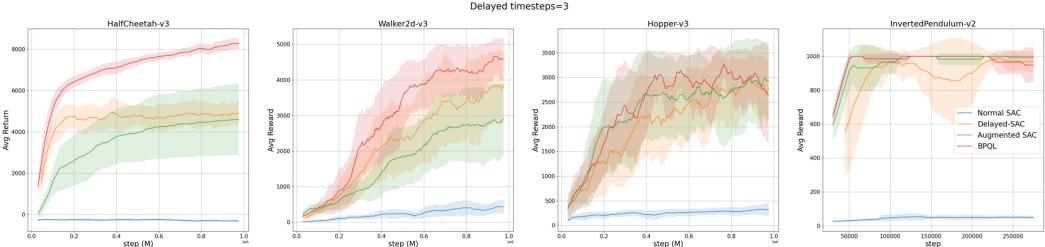

Figure 9: Performance curves for each algorithm in continuous tasks with 3 action delayed timesteps. We repeated the test for the tasks five times with different random seeds.

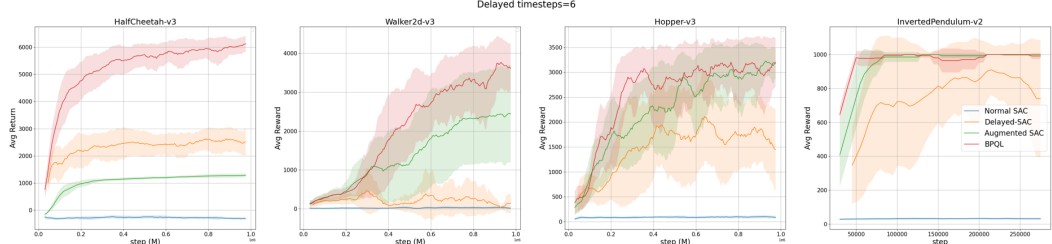

Figure 10: Performance curves for each algorithm in continuous tasks with 6 action delayed timesteps. We repeated the test for the tasks five times with different random seeds.

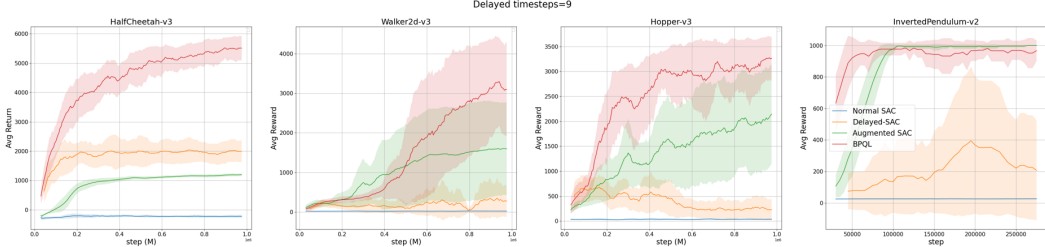

Figure 11: Performance curves for each algorithm in continuous tasks with 9 action delayed timesteps. We repeated the test for the tasks five times with different random seeds.

### F.3 Discrete Domain

We conducted additional experiments on the classical discrete control OpenAI gym [7] tasks: CartPole-v1 and LunarLander-v2. In the CartPole-v1 environment, there are two available actions, while in the LunarLander-v2 environment, four actions are available. We compared the performance of BPQL with the Delayed-Q algorithm [10] and Delayed-SAC in various action delay settings. To adapt BPQL and Delayed-SAC to the discrete setting, we implemented a discrete version of SAC [9] for both of them, instead of the original continuous version. We trained the agent for 1 million timesteps in all environments (except for Delayed-SAC and BPQL in the CartPole-v1 task, which was trained for 0.5 million timesteps), and obtained results using 10 random seeds.

The results showed that all algorithms performed similarly in the CartPole-v1 environment, which is considered relatively easier. However, in the LunarLander-v2 environment, as the delay increased, both Delayed-Q and Delayed-SAC showed a notable deterioration in performance. They even failed to control the agent in many episodes under the 10 timesteps delay setting. In contrast, the proposed BPQL algorithm consistently maintained high performance across all tested delay settings (3,5, and 10).

Table 4: Results of OpenAI Gym's discrete tasks. All results are shown with the standard deviation over the ten trials denoted by $\pm$. The scores of Delayed-Q in CartPole-v1 are sourced from the original paper [10] and marked with *.

| Environment | Delayed-Q [10] | Delayed-SAC (discrete ver.) | BPQL (discrete ver.) |
|---|---|---|---|
| CartPole-v1 (d=15) | 414±14* | 411±75 | 464±60 |
| CartPole-v1 (d=25) | 324±7* | 396±67 | 367±94 |
| LunarLander-v2 (d=3) | 247±23 | 285±3 | 275±10 |
| LunarLander-v2 (d=5) | 173±93 | 282±4 | 256±53 |
| LunarLander-v2 (d=10) | -84±67 | -72±178 | 245±59 |

