# OpenReview forum: "Belief Projection-Based Reinforcement Learning for Environments with Delayed Feedback"
_NeurIPS.cc/2023/Conference — NeurIPS 2023 poster_

### Official Review · Reviewer_ozYa · 2023-07-06

**Soundness:** 3 good
**Presentation:** 2 fair
**Contribution:** 2 fair
**Rating:** 6
**Confidence:** 2

**Summary:**

This work proposes a solution to the delayed feedback problem, in this case a fixed timestep delay. There are approaches to solving this problem in the literature, often associated with cumulative error or un-traceability given longer delays. One common approach — using augmented states by concatenating observed states & actions taken since visiting the delayed state — which grows exponentially the augmented state space as the delay increases. The authors’ algorithm (Belief Projection Based Q-Learning, BPQL) is formally shown to equally represent augmented state-based (Q-)Values. Further, the introduced belief-projection of those values is shown to be decomposable over the original states + residuals, therefore working on the original state-dimensionality rather than the exponential one of augmented state learning. The values are shown to be linearly approximable, an insight, which the authors then use to reformulate the discrete version into a continuous control version of a Belief-based Actor-Critic Algorithm, which outperforms naive/ augmented (-model) approaches of the soft-actor-critic in an delayed version of the MuJoCo benchmark.

**Strengths:**

- The approach is formulated on a quite technical level and appears to be a novel application of belief-based ML to the delayed feedback problem. There is related work in different directions cited for solving delayed environments (augmented approaches, model-based predictions, expectation minimization Q-Learning), although I am not very familiar with the augmented techniques, with are directly compared against here. The reasoning towards the belief based Q-Learning Actor-Critic seems to differentiate enough from Agarwal&Aggarwal2021, which they cite as closest related work.
- The mathematical reasoning seems reasonable as far as I can follow. The underlying constant delayed MDP and the assumptions are stated clearly. Furthermore, MuJoCo represents an appropriately difficult evaluation domain and the evaluation is sound, although there is no compute resource or training-time mentioned (as far as I can tell).

**Weaknesses:**

- The paper seems to be missing a discussion on the related concept of belief-state architectures, which may be considered related to the model-based approaches mentioned here but with more relevance to this chosen approach.
- One point of concern I have with the claimed novelty compared to Agarwal&Aggarwal2021 in the difference of developing a continuous control algorithm, rather than a discrete model. I find it confusing that much of the definition of the BPQL / Actor-Critic model revolves around Q- Values and Q-Learning (the Q-Function, as it is called here) which are an inherently discrete concept The authors mention a generalization to continuous control problems, but the transfer l.228-224 reads more like an afterthought to position the algorithm in a niche without direct comparison benchmarks. If most of the work covers the Q-Learning approach it would be informative to proof performance in this approach compared to e.g. Agarwal&Aggarwal2021 or the DDQN Model-based approach of van Hasselt. et. al. 2016 before evading into the novelty of continuous problems.
- Furthermore, the paper is lacking a motivation on why belief-based projection is an appropriate technique and was chosen here (aside from not dealing with exponentially growing state-spaces, which is repeated ... often enough). The reader is presented with a very final formulation of the belief matrix and belief projection operator initially, with the following pages re-contextualizing the known MDP / Bellman Equation into this framework. Without a background in control theory or kernel operators (and given the lack of references) I assume this part to be proposed by the authors or established theory, but without any descriptive intuition behind this formalization, much Chapter 3 reads like a proof that would be usually found in the Appendix. For a clearly readable publication I find this submission is written without the reasoning to convey its key-ideas behind the formalities to a broader audience.
- Given that the evaluation is based on the AC method, I would have preferred to have the AC technicalities in the paper itself rather than the Appendix. Some of the implementation details I found helpful to reason about the evaluation results were found in the Appendix and only the code itself. However, given the technical nature of the paper I would understand why they were cut.
-
I feel conflicted about claiming significant performance gains against the mostly self-constructed SAC baselines with ‘only’ 1M training steps on a complex domain like MuJoCo. The evaluation does seem cleanly run and favorable for the BPQL approach (at least in terms of sample efficiency), however, I would argue that only 1M steps is a debatable length for some of these problems to be solvable by the intentionally slow SAC algorithm (regardless of hidden-dim size, not even accounting for delay complexity). Out of the three baselines, the naive SAC is akin to the random baseline and the adjusted model-DDQN baseline was not originally designed for a SAC and is therefore hard to judge in terms of comparability. As mentioned above, it seems as if this work is introducing a Q-Value approach and evaluating on continuous control problems that are a short transfer away, but is not comparing to established discrete solutions to this problem. If better benchmarks for this specific environment niche (continuous control delayed feedback) are not available, it would be nice to see the theoretical maximum reward per domain or other relational metrics that would indicate how the BPQL is performing in general, not just compared to the benchmarks. Apart from the baseline evaluation, the belief-projection application seems like an improvement in terms of computability and therefore, sample efficient training for delayed environments.
- While the BPQL approach may be employed to any fixed step delay environment, with the initial setting of ‘uncertain latency’ that is motivated in the introduction, the authors are correct in listing this fixed-step assumption as a notable limitation. Especially in the continuous domain case (which is the differing focus of this work), having one (small) continuous, fixed delay signal is quite the niche. Nevertheless, given the defined setting the proposed algorithm shows promise in it’s intended purpose.


**Questions:**

Aside from concerns mentioned above, the following questions arise:
- Why is the belief-projection chosen here and how would you motivate this decision?
- Why was the SAC algorithm chosen as the baseline, how was it parameterized and how would common baselines like A2C or PPO compare in this setting?
- What is the runtime overhead of computing the belief-projection compared to the (technically simpler) augmented state techniques?
- Would it be possible to show e.g. memory-complexity of the baselines and of BPQL as a metric in the specific case of MuJoCo domains here? L.225ff mentions that the algorithm also stores augmented states (original states and action history) in an extra temporary buffer, could you please contextualize this to augmented state learning? (As mentioned above, this would also be information to include in the compute-resource / runtime statement.)

**Limitations:**

Assumptions and limitations are stated in the context of the formalization. No potential negative societal impact.

---

> ### Author Rebuttal · Authors · 2023-08-06
>
> Dear Reviewer ozYa,
>
> We want to sincerely thank you for the valuable effort you put into providing constructive feedback. We have carefully considered each of your comments, and provide additional details as follows:
>
> **Novelty**
>
> In addition to its applicability in continuous domains, BPQL has several distinct aspects compared to Agarwal&Aggarwal 's method (EMQL). In BPQL, the Q-function is evaluated using a reduced state size, while its policy uses an augmented state size to choose actions. Thus, the state sizes for the policy and Q-function are different, a **unique** feature not found in pure Q-Learning approaches like EMQL. Additionally, EMQL relies on a count-based model, making it challenging to apply in cases with large state and action dimensions, even in discrete environments. However, BPQL overcomes this issue by **not relying on an explicit model**, making it a truly "model-free" approach, in contrast to EMQL's "model-based" approach.
>
> **Discrete environments**
> >"it would be informative to proof performance in this approach compared to e.g. Agarwal&Aggarwal2021 or the Model-based approach ... before evading into the novelty of continuous problems."
>
> In the field of _robot learning_, latency is well-known for degrading performance. To address this issue, we developed BPQL, specifically designed for continuous robotic domains. Nevertheless, we agree that comparing our algorithm's performance with other algorithms in discrete domains would effectively demonstrate the generality of our approach. Therefore, we conducted an additional experiment with EMQL on discrete environments in OpenAI Gym. For this experiment, we implemented BPQL using the discrete version of SAC . The results are as follows:
>
> |CartPole-v0|d=2 / Epi=200|d=2 / Epi=1000|d=4 / Epi=200|d=4 / Epi=1000|
> |-|-|-|-|-|
> |EMQL|<= 80|<=150 |<=70|<=125|
> |BPQL|169 ±13|200±0|109±8|200±0|
>
> We also conducted another experiment to compare the performance with Delayed-Q (Derman et al.,2021) on discrete environments.
>
> |Cartpole-v1|Delayed-Q|BPQL|
> |-|-|-|
> |d=15|414±14|**464±60**|
> |d=25|324±7|**367±94**|
> |**Acrobot-v1**|**Delayed-Q**|**BPQL**|
> |d=15|**-211±53**|-246.8±206|
> |d=25|-351±57|**-335.3±202**|
>
> (_± denotes std._)
>
> BPQL's strong performance demonstrates its potential not only in continuous but also in discrete settings.
>
> **Longer Interaction**
> >“However, I would argue that only 1M steps is a debatable length ... ”
>
> To address this, we extended the evaluation to 2M and 3M steps. Model-based SAC didn't improve beyond 1M steps, but Augmented SAC showed notable improvement with more interactions due to its theoretical error-free TD learning. However, BPQL still outperformed it.
>
> |d=9|HalfCheetah-v3||Hopper-v3||Walker2d-v3||
> |-|-|-|-|-|-|-|
> |steps|2M|3M|2M|3M|2M|3M|
> |Augmented SAC|1362±40|1463±51|2476±783|2821±627|2970±897|3223±841
> |BPQL|5980±321|6214±108|2983±834|3120±561|4368±701|5130±472|
>
>
> **Relational metric**
> >“It would be nice to see the theoretical maximum reward per domain or other relational metrics ...”
>
> We agree that adding a relational metric in the comparison table would improve the clarity of our experiment and provide a better understanding of the impact of delays on the algorithms. Hence, we examined the performance of SAC in a "delay-free" environment as a comparison metric. The performance of delay-free SAC is as follows:
>
> |(1M steps)|HalfCheetah-v3|Walker2d-v3|Hopper-v3|Swimmer-v3|InvertedPendulum-v2| Reacher-v2|
> |-|-|-|-|-|-|-|
> |SAC|11202±423|4733±728|3145±429|91.2±12|1000±0.0|-3.8±0.6|
>
> **Questions**
> >**Q 1.** Why is the belief-projection chosen ... motivate this decision?
>
> We chose belief projection (BP) to find the best latent space representing the augmented state. In ML literature, dimensionality reduction methods like PCA and VAE have shown significant benefits for training. To leverage these advantages in RL, we incorporated a projection-based state reduction method into our algorithm. Additionally, as we discussed in Line 134-137, we assumed that similar successive transition probabilities imply similar meanings for the augmented states. Thus, we selected BP as the state reduction method.
> >**Q 2.** Why was the SAC algorithm ... compare in this setting?
>
>  SAC is one of the SOTA algorithms used in continuous tasks, usually outperforming PPO in the MuJoCo control tasks (Haarnoja et al. 2018). Therefore, we integrated SAC into BPQL. And the policy is parameterized to represent the squashed Gaussian distribution.
> >**Q 3.** What is the runtime overhead ... ?
>
> The dominant part of both algorithms in the runtime is the learning process (GPU load + neural network training).  We evaluated the Monte Carlo estimation for the learning process:
>
> ||Augmented SAC|BPQL|
> |-|-|-|
> |Learning time (1 iter.)|8.98 ms|9.32 ms|
> |||
>
> with the following resources:
> * CPU: i7 9700K
> * GPU: GTX 1080 TI
> * RAM: 48GB
>
> The experiments were conducted in the HalfCheetah-v3 with a delay of 9. BPQL uses a Q-function network with fewer parameters than Augmented SAC, yet it incurs 3% runtime overhead. This is likely due to the increased loading of transition tuples to the GPU from BPQL's temporary buffer.
> >**Q4**. Would it be possible to show e.g. memory-complexity ...
>
> Let M be the maximum size of the replay memory, A be the dimension of the action space, d be the delayed timesteps, and S be the dimension of the state space. Then the memory-complexity for Normal SAC is $O(M(A+2S+2))$, and for Augmented SAC, it is $O(M(A+2(\underbrace{S+dA}_{\text{augmented state}})+2)+Ad)=O(M(A(1+2d)+2(1+S)))$. The terms $Ad$ represent the space required for storing action history to create augmented states. BPQL adds two original-sized states to the tuple used in the replay memory of Augmented SAC, resulting in a complexity of $O(M⦁(A+2S+2(S+dA)+2)+2(S+Ad+1))=O(M(A(1+2d)+2(1+2S)))$.  The term $2(S+Ad+1)$ represents the space required for constructing the temporary buffers. Both are omitted in the Big O notation as M >> Ad and M >> 2(S+Ad+1).

---

> > ### Comment · Reviewer_ozYa · 2023-08-18
> > **Thank you for the rebuttal.**
> >
> > Thank you very much for this extensive reply and for taking the time to run these additional experiments. The additional insights regarding the the performance of BPQL in the discrete setting and the comparison to other works make a nice argument in favor of this work and will be taken in consideration. I hope these new results are included in the paper in some form. I also agree with the other reviewer and would very much like to see the code open-sourced for the final publication. I will adjust the review accordingly.

---

> > > ### Author Response · Authors · 2023-08-19
> > > **Thank you for the constructive suggestion!**
> > >
> > > We deeply appreciate the reviewer for providing the constructive suggestion and updating the score!
> > >
> > > We agree that the experimental results of the discrete environment will further support the promise of the proposed algorithm. Therefore, we will incorporate these experimental results into our paper.  Additionally, it's our pleasure to share our source code, and we look forward to our work contributing to the RL and robot learning communities. Thank you again for the valuable suggestion and for putting effort into reviewing our paper.

---

### Official Review · Reviewer_iPGm · 2023-07-08

**Soundness:** 3 good
**Presentation:** 3 good
**Contribution:** 3 good
**Rating:** 7
**Confidence:** 3

**Summary:**

The authors propose a projection approach for more compactly representing the value function of a delayed system in reinforcement learning. A complete algorithm based on SAC with neural network approximations is presented and benchmarked on four Mujoco environments where it outperforms competing approaches.

**Strengths:**

- The paper addresses an important problem when applying RL on real-world systems: latency. This is a common issue in robotics in particular. As such, a good solution would be of considerable value to the robot learning community.

- The paper is well-written and mostly easy to read

- The approach demonstrates impressive improvements on delayed systems in Mujoco benchmarks



**Weaknesses:**

- The experiments are only on four rather simple Mujoco control benchmarks. It would have been nice to have something more realistic, but I think the results were clear enough to establish the promise of the approach.

- A few important details could be more clear (see questions below)

Side note: Code is included but not publicly released. As the RL community has struggled with reproducibility due to small discrepancies in implementations, this would have been nice.

Minor:
- Not sure what you mean by "agnostic random delayed environment" in the conclusions? If you just mean randomly delayed environments, that sentence can be simplified a lot.

------------
After rebuttal:

The authors addressed all of my concerns and therefore I raise my score to a 7. This paper could be very useful to better solve delayed control problems, which I can attest to being common in robotics. That said I agree with Reviewer 3qfv that the method sections were needlessly difficult to read. However, the authors promised to add some intuitions earlier in the final manuscript, which I think will be good enough. I thank the authors for their work and encourage them to open source their code so there is less room for misunderstandings by future work on this important problem.



**Questions:**

1) The text is a bit vague on what added assumptions / complexity there is of the beta q-function representation. Initially it seemed like the state transitions/beta matrix would have to be estimated separately in (10) where a linear projection was used.  Later a "practical approximation" is introduced where it seems the beta Q-function can be updated iteratively via sampling as normal in Q-learning? Does this preserve optimality or is it an approximation (further approximation on top of regular NN SAC). If not optimal, can you clarify what added assumptions there actually are?

2) Why is it performing better even on the noisy environment without delay? This is not obvious from the claimed contributions.


**Limitations:**

To be clarified, the method seems very general.

---

> ### Author Rebuttal · Authors · 2023-08-06
>
> Dear Reviewer iPGm,
>
> Thank you very much for your positive review and for acknowledging our contributions. We have carefully read your comments, and we would like to provide additional details as follows:
>
> **Additional experiments on a realistic control task**
> >“The experiments are only on four rather simple Mujoco control benchmarks. It would have been nice to have something more realistic, but I think the results were clear enough to establish the promise of the approach”
>
> In addition to our experimental results, we performed an industry-oriented simulation task _Pusher-v4_ ;a collaborative robot to further validate the practicality of our algorithm. "Pusher-v4" is an environment from OpenAI Gymnasium, which features a multi-jointed robot arm resembling a human arm. The goal is to move a target object to a specified goal position using the robot's end effector.
>
> We conducted experiments in the Pusher-v4 environment where 200 ms and 500 ms sensing delay exist. The results are shown in the table below.
>
> ||delay=200 ms|delay=500 ms|
> |-|-|-|
> |Baseline (SAC w/o delay)|-21.0+-0.6|-21.0+-0.6|
> |SAC|-88.5+-10.9|-166.0+-3.9|
> |BPQL|-22.7+-0.7|-24.8+-1.5|
> ||||
>
> The results confirm the promise of BPQL, showing comparable performance to the delay-free environment (baseline SAC) despite the presence of delay.
>
> **Algorithm details**
> >“The text is a bit vague on what added assumptions ... Initially it seemed like the state transitions/beta matrix would have to be estimated separately in (10) where a linear projection was used.”
>
> Yes, In the practical version, we do not estimate the belief matrix and transition probability separately to compute the beta-Q value. Instead, we incrementally estimate the beta-Q-values based on the samples obtained through interactions with the environment. This sampling-based learning is a key characteristic of reinforcement learning, and in this setting, Q-values are proven to converge (Tsitsiklis and Van Roy, 1996).
>
> >“Later a "practical approximation" is introduced where it seems the beta Q-function can be updated iteratively via sampling as normal in Q-learning? “
>
> Yes. The beta Q-function learns similarly to a normal Q-function, using "timely properly-aligned" transition tuples $(s_t, a_t, r_t, s_{t+1}). $ However, this is **not same situation**. Because the action $a_t$ is obtained from the policy with the augmented input ($s_{t-d}, a_{t-d}, ... , a_{t-1}$) but not $s_t$ (which is not observed yet due to the delay). Consequently, unlike the normal Q-function that learns true Q-values, the beta Q-values approximates (linearly) the true Q-value, and BPQL has a unique characteristic: the (beta) Q-function is evaluated using a reduced (original) size of state, while its policy uses an augmented state size to choose an action (Eq. 24, 25).
>
> **Assumption behind the algorithm**
> >“Does this preserve optimality or is it an approximation ... If not optimal, can you clarify what added assumptions there actually are?”
>
> Unfortunately, neither the iterative method (Section 3) nor the practical version (Section 4) can guarantee optimality, even though the beta value converges. This is because the beta value is an approximation of the true value that ignores the residual (Eq. 11, 12). If an environment is extremely chaotic, the norm of this residual cannot be ignored smoothly, leading to a gap between the optimal policy and the learned policy.
>
> In this paper, we **assume** that the linearly approximated value in most real-world continuous tasks can **sufficiently represent** the true value. Indeed, in various MuJoCo benchmark experiments, BPQL showed better performance than the baselines and even outperformed Augmented SAC, which theoretically learns the true "error-free" value.
>
> **Optimality Analysis**
>
> Additionally, using a property of linear approximator, an interesting error-bound can be derived as follows:
>
> *Proposition.*
>
> Let  $\hat{V}^{*}_{proj.}$ be the unique fixed point calculated by repeatedly applying the combined operator
>
> $\Pi_{\textbf{W}}\bar{\textit{T}}^{\bar{\pi}}$, and $V_{true}$ be the true value of the policy. Then,
>
> $||V^{true}-\hat{V}^{\ast}_{proj.}||^2_W \leq \frac{1}{\sqrt{1-\gamma^{2}}}||V^{true}-\Pi_W V^{true} ||^2_W$.
>
> *Proof.*
>
> $||V^{true}-\hat{V}^{\ast}_{proj.}||^2_W$
>
> $=||V^{true}-\Pi_W V^{true}||^2_W$ $+||\Pi_W V^{true} -\hat{V}^{\ast}_{proj} ||^2_W$
>
> $=||V^{true}-\Pi_W V^{true}||^2_W+||\Pi_W V^{true}-\Pi_{W}\bar{T}^{\bar{\pi}}\hat{V}^{*}_{proj.}||^2_W$
>
> ($\because \hat{V}^{\ast}_{proj.}$ is the fixed point of the combined operator.)
>
> $\leq ||V^{true}-\Pi_W V^{true}||^2_W + \gamma^2||V^{true}-\hat{V}^{*}_{proj.}||^2_W$
>
> (by $\gamma$-contraction property of the combined operator.)
>
> $\blacksquare$
>
> This implies that the projection $\hat{V}^{\ast}_{proj,}$ is, at least, not far from the true value $V^{true}$.
>
> **Noisy Environment**
> >“Why is it performing better even on the noisy environment without delay”?
>
> In the noisy inverted pendulum environment, the delayed timesteps are six, not zero. We speculate that the reason for your misconception might be due to the small font size in the figure. If that's the case, we sincerely apologize for the confusion, and the font size will be adjusted for better readability in our camera-ready version of the paper.
>
> **Open-sourcing the code**
> >“Code is included but not publicly released. As the RL community has struggled with reproducibility ... this would have been nice.”
>
> Of course! We are thrilled and delighted to share our code to RL and robot learning community. Once our paper is published, the source code will be made accessible to everyone.
>
> **Minor**
> >“Not sure what you mean by "agnostic random delayed environment" in the conclusions? If you just mean randomly delayed environments, that sentence can be simplified a lot.”
>
> Thank you for pointing this out. We agree that removing 'agnostic' from the sentence simplifies it significantly. we'll make the modification in the our final version of the paper.

---

> > ### Comment · Reviewer_iPGm · 2023-08-13
> > **Thanks for the additional details**
> >
> > I thank the authors for the additional details which resolved most issues.
> >
> > I am happy to underscore that this is an important problem, delayed continuous domains are common in real-world control applications.
> >
> > My only remaining concern, that I see was also raised by some of the other reviewers, is with the clarity of the final steps from the initial linear projection idea to the sampling-based implementation. You first introduce a closed-form linear projection but then end up with (24) which looks like regular SAC, except for the augmented states in the expectation. However, the the augmented states are not used in the actual Q-functions, only for the policy entropy term (which is related to exploration). The text also mentions "parameterized" beta Q-functions for the first time which is confusing and hints that maybe there is something more going in the Q_beta value updates than is obvious from (24).

---

> > > ### Author Response · Authors · 2023-08-14
> > > **Additional details for Equation (24).**
> > >
> > > We sincerely appreciate your enthusiastic engagement in the discussion! We hope that the derivation of Equation (24) becomes clearer through the following details.
> > >
> > > **step 1. Regular Bellman backup**
> > >
> > > First, Bellman backup for regular Q-learning can be expressed as the following equation:
> > > $Q(\bar{s_t}, a_t)=R(\bar{s_t}, a_t)+\gamma  E_{P(\bar{s_{t+1}}|\bar{s_t},a_t), \bar{\pi}(a_{t+1}|\bar{s_{t+1}})} [Q(\bar{s_{t+1}},a_{t+1})].   \qquad (1)$
> > >
> > > **step 2. Bellman backup with linear approximation**
> > >
> > > If we approximate this Q-function using a linear function approximator, Equation (1) can be transformed as follows:
> > >
> > > $\beta^T\Phi(\bar{s_t},a_t)  =R(\bar{s_t}, a_t)+\gamma E_{P(\bar{s_{t+1}}|\bar{s_t},a_t), \bar{\pi}(a_{t+1}|\bar{s_{t+1}})}[\beta^T\Phi(\bar{s_{t+1}},a_{t+1})], \qquad (2)$
> > >
> > > where $\Phi$ is the feature vector and the $\beta$ is the corresponding parameters. Now our object is to find the parameters $\beta$ that best fit Equation (2) for the feature vector $\Phi$.
> > >
> > > **step 3. Belief projection-based linear approximation**
> > >
> > > In the case of belief projection, $\Phi$ equals $[P(s_1|\bar{s_t}), P(s_2|\bar{s_t}), ... , P(s_{|\chi|}|\bar{s_t})]^T$ and $\beta$ equals $[Q_{\beta}(s_1,a_t),Q_{\beta}(s_2,a_t), ... ,Q_{\beta}(s_{|\chi|},a_t)]^T.$ Therefore, Equation (2) can be rewritten as the following:
> > >
> > >
> > >
> > > $E_{P(s_t|\bar{s_t})}[Q_{\beta}(s_t,a_t)]=R(\bar{s_t}, a_t)$+$\gamma E_{P(\bar{s_{t+1}}|\bar{s_t},a_t), \bar{\pi}(a_{t+1}|\bar{s_{t+1}})}$ $[E_{P(s_{t+1}|\bar{s_{t+1}})}[Q_{\beta}(s_{t+1},a_{t+1})]].  \qquad (3)$
> > >
> > > To close the gap between the left and right sides of Equation (3), we minimize the following equation:
> > >
> > > $E_{P(s_t,s_{t+1},a_{t+1},\bar{s_{t+1}}|\bar{s_t})}[Q_{\beta}(s_t,a_t)-\bar{R}(\bar{s_t},a_t)-\gamma Q_{\beta}(s_{t+1},a_{t+1})]^2. \qquad (4)$
> > >
> > > ($\because E_{P(s_t,s_{t+1},a_{t+1},\bar{s_{t+1}}|\bar{s_t})}Q_{\beta}(s_t,a_t))=E_{P(s_t|\bar{s_t})}[Q_{\beta}(s_t,a_t)]$)
> > >
> > > This linearly-approximated Bellman backup (Equation 4.) should hold for all states, so our final objective can be expressed as the following:
> > >
> > > $E_{\bar{s_t} \sim \rho(\bar{s})}  [E_{P(s_t,s_{t+1},a_{t+1},\bar{s_{t+1}}|\bar{s_t})}[Q_{\beta}(s_t,a_t)-\bar{R}(\bar{s_t},a_t)-\gamma Q_{\beta}(s_{t+1},a_{t+1})]]^2. \qquad (5)$
> > >
> > > We utilize replay memory to compute Equation (5). The replay memory contains tuples, where each tuple represents a transition. The transition  includes the agent's action $a_t$, the subsequent states $\bar{s_{t+1}}, s_{t+1}$, and the reward $R(\bar{s}_t,a_t)=r_t$ when
> > >  $\bar{s_t}$ is given.
> > >
> > > Therefore, we approximate Equation (5) as follows using the replay memory $D$:
> > >
> > > $E_{(\bar{s_{t+1}},s_t,r_t,\bar{s_{t+1}},a_{t+1},s_{t+1},\bar{s_t})\sim D}[Q_{\beta}(s_t,a_t)-\bar{R}(\bar{s_t},a_t)-\gamma Q_{\beta}(s_{t+1},a_{t+1})]^2. \qquad (6)$
> > >
> > >
> > > (_Please note that the beta function;_  $\beta(s,a)=Q_{\beta}(s,a)$ _differs from the conventional Q-value in RL literature, yet the expectation of the beta Q-value is the conventional Q-value. However, given the similarity of the beta function update equation to the regular Q-learning formula, we have specifically chosen to name the  beta function as_ $Q_{\beta}(s,a)$).
> > >
> > > **step 4. Continuous domain**
> > >
> > > Lastly, for the application of Equation (6) in continuous domains, we have parameterized the function $Q_{\beta}(s,a) \to Q_{\beta, \theta}(s,a)$ using a neural network's weights $\theta$. If we use soft Bellman backup instead of Bellman backup, the Equation (24) in the manuscript is established.
> > >
> > > This aligns with our intuition: replacing the last observed state and the previous $d$ actions (forming an augmented state) with the state $d$ steps ahead (the result of the previous actions and the last observed state).
> > >
> > > Please note that Equation (6) looks similar to regular Q-learning, but it has distinct differences. Because, in the delay setting, ignoring the delay can lead to a violation of the Markovian assumption in the one-step transition; $P(s_{t+1}|s_t,a_t) \neq P(s_{t+1}|s_{t},a_{t},a_{t-1}, ...)$. Also, the action $a_t$ is chosen from $\bar{\pi}(\cdot|\bar{s_t})$, not $\bar{\pi}(\cdot|s_t)$.

---

> > > > ### Comment · Reviewer_iPGm · 2023-08-14
> > > > **Thanks for the clarifications.**
> > > >
> > > > Thanks! It's much clearer to me now. I think the reason I found it difficult to read is because the workings of the idea is only presented piecemeal through the math, and there are a lot of steps in the paper. I feel an outline of the solution could have been given somewhere earlier on.
> > > >
> > > > For example, I think the gist of it is something like this:
> > > > You propose to do the Q-updates in the original, smaller (non-Markovian) state-space using a linear ("belief") projection down from the larger action-augmented Markovian state.
> > > > 1) This does not violate optimality of Bellman updates since it is a linear approximation.
> > > > 2) The linear projection does not have to be explicitly computed as it can be rewritten as an expectation (which is a linear operator). You can sample the solution to this as in regular Q-learning, just using a replay buffer with the augmented state with the extra delayed actions.
> > > >
> > > > Minor: Under Eq. 7 you write "where d denotes the number of delayed timesteps in the environment.", but I cannot see a d in the equation. Is this a typo?
> > > >
> > > > The reason "parameterized beta Q_beta,theta" confused me is because in continuous control, Q is pretty much always parameterized and it was unclear if the "parameterized" referred to theta or something going on with beta. You could maybe hint that these are just the regular NN theta parameters.

---

> > > > > ### Author Response · Authors · 2023-08-15
> > > > > **Thanks for the constructive feedback.**
> > > > >
> > > > > We truly appreciate the very valuable and constructive feedback you have provided, and we are happy to see that it's now much clearer to you!
> > > > >
> > > > > We agree that providing an outline of the solution before deriving the equations in Section 4 will improve the reader's understanding of the algorithm even further. Therefore, we will briefly summarize the followings at the beginning of Section 4 in the revised version of the paper.
> > > > >
> > > > > * We will compute Q-values in the smaller state space created through belief projection, rather than in the large augmented state space.
> > > > > * The approximation of Q-values through belief projection is a linear approximation, which still guarantees the convergence of the Bellman equation.
> > > > > * The solution can be achieved through a sampling-based RL framework without explicitly constructing the belief matrix.
> > > > >
> > > > > Additionally, it is noted that the first and second subscripts of $Q_{\beta, \theta}(\cdot, \cdot)$ refer to the beta Q-values and the neural network weights, respectively. Actually, the beta Q-values depend on the neural network weights. Nevertheless, $\theta$ and $\beta$ are written together for clearer notation. Only $\theta$ is a variable set to be optimaly taken through learning. We will ensure to make this part clear in the revised version of the paper. We sincerely appreciate your feedback!
> > > > >
> > > > > **Minor typo**
> > > > >
> > > > > Yes, there is a typo. "$\forall t > d$" needs to be included in Equation (7). Thank you for carefully reading our paper and pointing out this!

---

### Official Review · Reviewer_HRKg · 2023-07-13

**Soundness:** 3 good
**Presentation:** 4 excellent
**Contribution:** 3 good
**Rating:** 7
**Confidence:** 3

**Summary:**

This paper aims to tackle the delayed feedback RL problem. In such a problem setting, usually there is some fixed/constant delay in the environment, so that the observed rewards are delayed by $d$ timesteps. Traditionally, augmented state spaces have been used to solve such an issue, however, for large delays this leads to an exponentially large dimension for learning the value function. Thus, the proposed approach (BPQL) leverages the deconstruction of the augmented value function via the belief matrix into a lower dimensional beta q value function, and shows that this approximates the augmented q function (derived from the soft bellman update). This outperforms baselines on a variety of control tasks.

**Strengths:**

- I think there is good technical novelty and soundness in this paper
- The problem space is very interesting and relevant to real world applications
- BPQL drastically outperforms basleines
- This approach is general (as it applies to continuous settings)

**Weaknesses:**

I think it would be interesting to explore this idea on different types of environment delay (for example delays coming from different sources like actuators, sensors, etc. It would also be good to see more analysis and ablations on different levels of delays (value of $d$).

Finally, it would be great if this apporach applied to more realistic robotics settings (even simulated versions).

**Questions:**

See weaknesses section

**Limitations:**

This has been sufficiently addressed

---

> ### Author Rebuttal · Authors · 2023-08-07
>
> Dear HRKg,
>
> We sincerely appreciate your positive review and your interest in the problem space our paper aims to address. Based on your suggestions, we have conducted additional tests and provide the details as follows:
>
> **Different types of delay & Ablation**
> >“I think it would be interesting to explore this idea on different types of environment delay (for example delays coming from different sources like actuators, sensors, etc. It would also be good to see more analysis and ablations on different levels of delays (value of d).”
>
> In the field of robotics, delays occur in various scenarios such as remote robot control, communication in low bandwidth, and complex control algorithm with on-board devices (sensors, actuators) with low computational capacity. Environments with these delays can be categorized into the following three types:
> * Observation delay environment
> * Action delay environment
> * Combined (Observation + Action) delay environment
>
> In the main text, BPQL mainly conducted experiments in environments where observation delay only exists. However, BPQL is a general algorithm that can be applied not only to observation delay scenario but also to action delay scenario and even the combined delay scenario. To validate the generality of BPQL, we conducted an additional experiment evaluating performance under various delay combinations, and the results are listed in the table below.
> ||HalfCheetah-v3|Walker2d-v3|Hopper-v3|
> |-|-|-|-|
> |Obs. delay=5 / Act. delay=0|6371.4+-181.9|4363.3+-332.0|3144.1+-528.17|
> |Obs. delay=0 / Act. delay=5|6459.1+-204.8|4008.2+-663.8|2736.8+-931.6|
> |Obs. delay=3 / Act. delay=2|6332.4+-210.4|4399.7+-455.5|2802.8+-670.8|
> |||||
>
> From the results, we confirmed that BPQL shows promise not only in single delay environments but also in combined delay environments. This generality of BPQL allows for broader application in addressing the significant latency issue in the field of robot learning.
>
> **Additional experiments on a realistic robotics setting**
> >"it would be great if this approach applied to more realistic robotics settings (even simulated versions)."
>
> To further demonstrate the practicality of BPQL, we conducted an additional experiment in a more industry-oriented simulation task called Pusher-v4 from OpenAI Gymnasium. Pusher-v4 is a collaborative arm robot with multiple joints, resembling a human arm. The objective of the task is to move a target object to a specific goal position using the robot's end effector. In this experiment, we added sensing delays of 200 ms and 500 ms in the environment, and the results are presented in the table below.
>
> ||delay=200 ms|delay=500 ms|
> |-|-|-|
> |Baseline (SAC w/o delay)|-21.0+-0.6|-21.0+-0.6|
> |SAC|-88.5+-10.9|-166.0+-3.9|
> |BPQL|-22.7+-0.7|-24.8+-1.5|
> ||||
>
> The results show that, even with the presence of delay, BPQL demonstrates comparable performance to the delay-free environment (baseline SAC), reaffirming its potential.

---

### Official Review · Reviewer_3qfv · 2023-08-09

**Soundness:** 2 fair
**Presentation:** 2 fair
**Contribution:** 2 fair
**Rating:** 4
**Confidence:** 1

**Summary:**

This paper addresses the issue of delayed feedback in reinforcement learning,
where the observations and rewards received by the agent are those generated by
the environment multiple (`d`) time steps ago, most commonly caused by latency
in hardware. This problem is typically solved by augmenting the policy and
value functions with the `d` previous actions, recovering the Markov property
of the problem. This paper argues that this augmented 'state space' is
prohibitively large and proposes Belief-Projection-Based Q-learning (BPQL) as a
fix to the issue.

BPQL modifies SAC such that the value function (critic) that is being learned
only takes the observation (and not the `d` actions) as input (though the
policy (actor) still depends on the augmented space). The result is an
approximation of the original problem, where the critic learns the expected
value of the policy based on just the last observation, as opposed to the
observation and last `d` actions.

This approach is motivated by a substantial theoretical section that derives
and discusses the (approximated) loss function. The method is compared against
augmented SAC, a model-based approach, and regular SAC on MuJoCo for different
delays.

**Strengths:**

This paper proposes a high-performance solution to the complex problem of
delayed feedback. It does so in a fairly simple (in the positive sense) way
that should be easily implementable and testable.

Another advantage is the theoretical support of what otherwise would be a
fairly ad-hoc solution. This is particularly useful because, in my opinion, the
intuition of BPQL is fairly nice: the critic can ignore the additional action
history input because this depends only on the policy it is evaluating: the
action sequence of the policy is relatively 'stable', or determined, so not
particularly variable or something that needs to be given to the critic in
order to evaluate the expected return.

======== post rebuttal ==========

The authors brought up reasonable arguments, context, and experiments to defend their work. I would personally still learn towards rejection due to the fact that, in its current form, I believe that the combination of the presentation, algorithmic contribution, and empirical evaluation are in my opinion relatively weak. However, I do not feel strong enough about this to champion for a rejection when others disagree.

**Weaknesses:**

Despite these strengths, there are a few concerns that makes me lean towards
rejection.

While the experiments show otherwise, I am slightly surprised that the action
space is big enough to cause issues in terms of policy evaluation and would
need more data to be truly convinced. Clearly augmented SAC is suffering but,
unless there is something particular about these problems (of which I have no
detailed knowledge), generally the observation/state space is so much larger
than the action space that the additional input of `d` actions should not be
that significant.

Regarding clarity, the presentation of the paper was in my opinion confusing:
First, the method section contained a substantial amount of information that
would fit better in the background (e.g. 3.1, 3.2, and the start of 4). Second,
there was a very limited amount of information on the idea behind the paper or
its overview until you start digging into the details. For example, until the
last paragraph before the experiment section I had no idea where this was
going. Additionally, the theoretical section seems to attempt to justify the
approximations made by BPQL, but that was not clear at all until diving into
the section afterwards.

Most importantly, i still struggle to understand the theoretical support. I
believe it eventually it provides a theoretical explanation for BPQL's loss
function, but I only truly understand the loss function from a intuitive
perspective and had to gloss over section 3 altogether. The main reason I lean
towards rejection is because it was not presented in a way that the theoretical
section was a contribution to me. This mostly because it is fairly 'easy' to
propose to 'just drop' some input to reduce the space and have it work well for
some problems. So the theory is important here but, in my perspective,
difficult to understand.

Lastly, some ablations that would be very informative are missing. For example,
it is clearly important that the policy accepts the augmented space (because
that seems to be the key difference between BPQL and 'normal SAC'), but the
obvious alternative is to do some variant where the policy takes in only the
last observation but the critic gets the augmented input. I think more in depth
experiments would further motivate the approach. In particular, I would want to
see why it is so important that the policy does accept the last `d` actions as
input: is this a property of the problems, or is there some theory or intuition
that explains this discrepancy?

**Questions:**

N/A

---

> ### Author Response · Authors · 2023-08-11
> **Response to emergency reviewer 3qfv**
>
> Dear 3qfv,
>
> We want to sincerely thank you for the effort you put into providing valuable comments, and provide additional details as follows:
>
> **Impact of additional input of actions**
> >"Clearly augmented SAC is suffering but, unless there is something particular about these problems (of which I have no detailed knowledge), ... the additional input of d actions should not be that significant."
>
> As you mentioned, the dimensions of the original state take up a substantial portion in the augmented state. For instance, in the Walker2d-v3 environment with a delay of 9, the augmented state is made up of 17 dimensions from the original state and 54 dimensions from the additional actions. (Note that even in this particular case, the ratio of dimensions influenced by the added actions is already **six times larger!**)
>
> However, we think that these two factors have different effects on increasing the size of the state space. Because, in the field of robot learning that we are focusing, components such as velocity and angular velocity which make up a robot's state are influenced by the actual environment and actuators. Consequently, each component's size is intrinsically limited. This is similar (though not identical) to the context of the Manifold Hypothesis in deep learning literature:  most high-dimensional data lies on low-dimensional manifolds.
>
> However, added actions are the problem. Because, the additional actions can be completely random, unlike the state space. For instance, in reinforcement learning, a policy tends to select actions randomly in the beginning to encourage exploration. This randomness becomes a factor that dramatically increases the space of the augmented state.
>
> To support this assumption, we conducted an experiment to investigat the difference in the convergence of the Augmented Q-function when using **1) a fixed untrained policy**  and **2) a fixed trained policy**. We compared the TD-error of the Q function when applying both policies, and the results are as follows.
>
> **Walker2d-v3 with delay 9**
> |steps|50k|100k|150k|200k|250k|
> |-|-|-|-|-|-|
> |TD-error of untrained policy|551.7|422.7|194.9|110.4|92.0|
> |TD-error of trained policy|57.0|57.5|56.4|56.3|57.7|
>
> The results showed that the TD-error converges rapidly in the trained policy, while the untrained policy shows a slower convergence speed. This demonstrates how the randomness of initial policy significantly affects Q-value convergence, thereby confirming that the additional input of actions has a significant impact on augmented Q-function convergence.
>
> **Contribution**
> >"Most importantly, i still struggle to understand the theoretical support ... the main reason I lean towards rejection is because it was not presented in a way that the theoretical section was a contribution to me. This mostly because it is fairly 'easy' to propose to 'just drop' some input to reduce the space and have it work well for some problems. So the theory is important here but, in my perspective, difficult to understand."
>
> As you mentioned, it is easy to intuitively grasp how our proposed algorithm works effectively from a practical perspective. However, through Section 3 (the theory section), we aimed to demonstrate why we chose belief-projection for reduction technique, and why using belief-projection to train the critic leads to stable convergence even though the beta Q-values approximate the true Q-values. We believe that explaining the underlying reasons behind the successful performance of an algorithm will provide **valuable guidance** to machine learning researchers interested in this field for their future studies.
>
> **Why we train the policy using augmented states.**
> >"In particular, I would want to see why it is so important that the policy does accept the last d actions as input: is this a property of the problems, or is there some theory or intuition that explains this discrepancy?"
>
> The idea of feeding a reduced sized state into the policy rather than an augmented sized state can be naturally considered. However, we respectfully state that, unfortunately, this is not feasible in BPOL. In an environment delayed by $d$ timesteps, the agent can only see the delayed observation $s_{t-d}$  at time $t$. In other words, unlike the learning stage of the critic, there is no way to know the $s_t$ corresponding to the augmented state $(s_{t-d}, a_{t-d+1}, ..., a_{t-1})$. Therefore, in BPQL, the policy uses the augmented state that contains the **most information at that point to make decisions.**  Note that depending only on the last observed state wouldn't be enough to make the right decision. This is because a series of actions significant impact on the current true state. Consider a game of chess, for example. The current true chess board state is heavily influenced by previous actions, making it challenging for the agent to decide a right action based solely on the last observed state. Hence, the policy should be trained with the augmented state.

---

> > ### Comment · Reviewer_3qfv · 2023-08-11
> > **Thank you for the detailed rebuttal**
> >
> > Thanks for the clear response. it has helped me understand the broader picture and will update my review accordingly.

---

> > > ### Author Response · Authors · 2023-08-15
> > > **RE: Thank you for the detailed rebuttal**
> > >
> > > We sincerely appreciate the reviewer's effort in reviewing our manuscript, providing constructive feedback, and raising his/her score!

---

### Official Review · Reviewer_MGmU · 2023-08-09

**Soundness:** 4 excellent
**Presentation:** 3 good
**Contribution:** 3 good
**Rating:** 7
**Confidence:** 3

**Summary:**

The paper proposes a new belief projection-based method for approximating delayed states in reinforcement learning. The method is derived from basic principles with properties of the projection demonstrated. The method is then transformed into a practical, model-free approach, that is extensively evaluated and bench-marked, with promising results.

**Strengths:**

- The paper is easy to follow, the derivations are cleanly done and well explained.
- The idea is interesting and sound
- I really liked the logical progress of deriving the method with the assumption of having access to all the probabilities etc. and then step by step turning it into a practical method
- The experiments show a good number of baselines, and sufficiently many standard benchmarks. The ablation with the delay steps, network capacity, and stochasticity are well done. The number of repetitions is adequate.
- The method shows clear experimental benefits

**Weaknesses:**

- I found Sect 2.1 and 2.2 a tiny bit confusing in terms of notation initially. Especially in Sect 2.1 it was initially not clear why X is used and initially it wasn't clear to me that the reward is also delayed (Sect 2.2). Just some minor writing tweaks
- Sect. 2.2: in conventional control having a system-model-based observer (e.g. Kalman filter) is also quite common - not just the augmented state - similar to the methods described a bit further down
- In Sect. 1 it almost sounds like the method also considers (or should consider) varying delays for e.g. hardware issues. I'd suggest making more clear from the start that you consider constant delays.
- I would have appreciated some little outline/forward pointer. On the first read I was a bit puzzled by the apparent assumption e.g. in Eq (7) of having access to the full model while initially a model-free method was promised. Briefly explaining the structure of the paper in the beginning would avoid that confusion.
- missing limitations (see below)

**Questions:**

- See small writing suggestions above
- The belief projection method comes a bit out of the blue, can you better motivate why this is a good idea?
- The setting is connected to POMDPs where there is quite a bit of literature but which is only mentioned in passing. Can you discuss the relation a bit better?
- Please detail the limitations better, you mention random delayed environments which are an obvious limitation.
  * Do you think the method would still perform reasonably if there is some small randomness in the delay (even if that's not backed by the theory)?
  * What are the implications of the linear approximation?
  * More generally, this seems to be a lossy compression, bringing everything back down to the original state-space size. In which cases will having the full augmented state (and infinite data etc.) still result in better performance?
  * Can you do something in between, i.e., project to a slightly larger state-space?

**Limitations:**

- societal impact is not really applicable
- an explicit discussion on limitations is a bit short (see above)

---

> ### Author Response · Authors · 2023-08-11
> **Response to emergency reviewer MGmU**
>
> Dear reviewer MGmU,
>
> We sincerely appreciate the reviewer for providing valuable feedback and showing interests in our manuscript. We have carefully considered each of your comments, and provide additional details as follows:
>
> **Motivation**
> >"The belief projection method comes a bit out of the blue, can you better motivate why this is a good idea?"
>
> Many machine learning references have shown that dimensionality reduction methods, such as variational autoencoders (VAE), contribute to efficiently training neural networks. To leverage these advantages in the RL framework, we concluded to use an effective reduction method for the augmented state space. As we discussed in Lines 134-137 of the main text, if two successive transition probabilities, $P(s_t|\bar{s_t^1})$ and $P(s_t|\bar{s_t^2})$, are similar, we can assume that the augmented states $s_t^1$ and $s_t^2$ share similar representative meanings. In other words, if two augmented states (created by a sequence of actions following the last observed state) and their corresponding current _true_ states are similar, we assumed that these two augmented states would possess similar representative meanings. This insight guided us to apply the belief projection as the dimensionality reduction method.
>
> **Delayed enviornment and POMDPs**
> >"The setting is connected to POMDPs where there is quite a bit of literature but which is only mentioned in passing. Can you discuss the relation a bit better?"
>
> Environments with delays are equivalent to partially observable MDP (POMDP) environments. If we train an agent in this POMDP while ignoring the delays, the agent will end up with a _suboptimal_ policy. Therefore, for the agent to obtain an optimal policy,  using the augmented state method to simplify the POMDP into an MDP, or using an explicit model for state estimation, is often practiced.
>
> **Questions**
> >"Do you think the method would still perform reasonably if there is some small randomness in the delay?"
>
> We conducted an additional experiment to investigate whether our proposed algorithm shows good performance even in an environment that includes small randomness. The experiment was conducted in the HalfCheetah-v3 environment, and the agent was trained in an environment with a constant delay of 3. The agent is evaluated in the following randomly delayed environment: The agent observes a 3 timesteps delayed state with a probability of 0.95, and the agent observes a randomly delayed state within the range [1, 5] with a probability of 0.05. The results are listed in the table below.
>
> ||baseline (constant delayed env.)|randomly delayed env.|
> |-|-|-|
> |Performance|8743+-103|7119+-628|
> ||||
>
> BPQL shows its **robust** performance in the presence of small randomness. However, there was small performance deterioration in the random delay environment.
>
> >"What are the implications of the linear approximation?"
>
> The implication of the linear approximation is that the augmented state-based Q function is represented as a linear combination of the original state-based Q function in lower dimensions, such that it has the closest weighted Euclidean distance. In other words, we used belief projection to express this augmented state-based Q function as an expected value of the beta-Q function.
>
> >"More generally, this seems to be a lossy compression ... In which cases will having the full augmented state (and infinite data etc.) still result in better performance?"
>
> Yes, our proposed algorithm BPQL uses a linear approximation method, which inevitably produces some residuals (Eq.11 and 12). In other words, as you mentioned, this is a lossy compression. So, in a simple environment (where the augmented state space isn't too large), the augmented approach might perform better. Because, this method does theoretically "error-free" temporal difference (TD) learning.
>
> For instance, let's consider the InvertedPendulum-v2 environment, where observations have a dimension of 4 and actions are of dimension 1. If the delayed timesteps is 6, the total dimension of the augmented state becomes only 4 + 1 * 6 = 10. In such a relatively simple environment, the Augmented SAC method demonstrates more stable convergence of performance as depicted in Figure E in Appendix section.
>
> >"Can you do something in between, i.e., project to a slightly larger state-space?"
>
> In the propoesed algorithm BPQL, this is not possible. Because BPQL approximates the augmented state in the _original_ state space, minimizing the weighted Euclidean norm between the approximated values and the actual values. However, we find this to be a very interesting question. In an extremely delayed environment, it may be hard to ignore the information loss arising from the linear approximation of belief projection. In such situations, using other **non-linear dimensionality reduction methods** such as VAE could be advantageous, **which can project to a slightly larger state-space.**

---

> > ### Comment · Reviewer_MGmU · 2023-08-11
> > **thanks for the rebuttal**
> >
> > Thanks a lot for the detailed replies and additional experiment. This helped me significantly to understand your method better.
> > My comment on POMDPs might have not been entirely clear: the connection in terms of setting is obvious, what I was looking for is relations to methods for solving (more general) POMDPs.

---

> > > ### Author Response · Authors · 2023-08-12
> > > **Response to emergency reviewer MGmU (2)**
> > >
> > > We are very delighted to see that your understanding of the manuscript has been clarified through the response!
> > > For the POMDPs part, we list several methods to address more general POMDPs as follows:
> > >
> > > **State Estimation Methods**
> > >
> > > To begin with, we have the Kalman Filter-based approach. For a linear stochastic state space model, the so called linear quadratic Gaussian(LQG) control theory has been well established, which optimizes the expected value of a given quadratic cost function. Remarkably, such a theoretically well-understood LQG control is separated into a deterministic linear quadratic(LQ) control and the Kalman filter. Much more generalized from the basic problem setting of LQG control, this paper considers nonlinear and model-free systems, realistic nonquadratic reward functions, and intractable time-delay. These issues are almost impossible to handle in classical model-based explicit approaches such as the above LQG control theory. Furthermore, the aforementioned separation principle does not hold. It means that Kalman filter is not optimal. This paper proposes a practical approach for controlling nonlinear systems with time-delay and more general non-quadratic reward functions in a data-driven manner, without requiring any prior model information. Specifically, the paper addresses efficient state exploration by compactly reducing the augmented state space size, as well as value-function representation for practical implementation.
> > >
> > > In addition, there are attempts to estimate states by learning the dynamics of the model in various ways. [Agarwal and Aggarwal] construct the dynamic of transitions based on count-based model. Furthermore, [Derman et al] utilize neural networks to learn the dynamic model of the environment. Moreover, an RNN-based architecture can also be used to build dynamic models [Matthew and Stone]. However, these approaches, though intuitive, tend to accumulate errors when predicting states that are far from the current timestep, consequently impeding agent learning. In this paper, we address this issue of model errors using a model-free approach rather than a model-based approach.
> > >
> > > **Complete Information Methods**
> > >
> > > In this method, states with incomplete information are transformed into states with complete information. It is one of the widely used approaches to solving POMDPs. [Mnih, Volodymyr, et al.] constructed a complete state by stacking past states, and [Simon Ramstedt and Christopher Pal] created states with complete information by adding a single action. The augmented approach discussed in this manuscript can also be categorized under the complete information methods. However, a limitation of this method is the exponential growth of the state space, which impedes the convergence of the value function.
> > >
> > > **Relevance to the existing POMDP approaches**
> > >
> > > This paper deals with a POMDP problem through an an approximated MDP approach, which basically serves to provide benefits of the proposed scheme. The proposed algorithm, BPQL makes an attempt to efficiently use the second method above for POMDP to MDP conversion. The explosive nature of increasing data dimensions in the second method is strategically addressed in BPQL by approximating the augmented state space to a lower dimension. As the delay size increases, the existing methods suffer from the curse of dimensionality. However, this is not the case with BPQL. The state dimension size of BPQL is constant without respect to the delay size, which is an unprecedented advantage. Since the POMDP method itself is not a main issue, its description was not detailed. Nevertheless, we agree that briefly summarizing the literature related to POMDPs would enhance broader reader comprehension. Therefore, we will include a more comprehensive discussion about POMDPs in our revised version of the paper. We sincerely appreciate your constructive suggestion!
> > >
> > > **Reference**
> > >
> > > [1] Agarwal, Mridul, and Vaneet Aggarwal. “Blind decision making: Reinforcement learning with delayed observations.” Pattern Recognition Letters 150 (2021): 176-182.
> > >
> > > [2]Derman, Esther, Gal Dalal, and Shie Mannor. “Acting in delayed environments with non-stationary markov policies.” arXiv preprint arXiv:2101.11992 (2021).
> > >
> > > [3]Hausknecht, Matthew, and Peter Stone. “Deep recurrent q-learning for partially observable mdps.” 2015 aaai fall symposium series. 2015.
> > >
> > > [4]Mnih, Volodymyr, et al. “Human-level control through deep reinforcement learning.” nature 518.7540 (2015): 529-533.
> > >
> > > [5]Ramstedt, Simon, and Chris Pal. “Real-time reinforcement learning.” Advances in neural information processing systems 32 (2019).

---

> > > > ### Comment · Reviewer_MGmU · 2023-08-14
> > > > **thanks**
> > > >
> > > > a lot for the very detailed response. If you mange to squeeze in a brief summary, that would be great!

---

> > > > > ### Author Response · Authors · 2023-08-15
> > > > > **A brief summary of the response**
> > > > >
> > > > > We sincerely appreciate your active engagement and the constructive suggestion!
> > > > >
> > > > > The detailed description above boiled down to the following compact statements
> > > > > in the revised manuscript:
> > > > >
> > > > > POMDP has been mainly handled with two approaches: the state estimation methods and the complete information methods. As inherent drawbacks, the first one requires accurate model-based state estimation and the second one suffers from the curse of dimensionality. So, the research objective of this work is to resolve these two drawbacks together, i.e., find a model-free approach that is basically free of dimensionality curses. This is why the proposed BPQL was born.

---

### Decision · Program_Chairs · 2023-09-21

**Decision:**

Accept (poster)

**Comment:**

The paper proposes a new method for solving decision making problems with delayed feedback. The approach is based on reducing the augmented state dimension by belief projection. The practical method derived from the theoretical results is extensively evaluated.

The reviewers found the proposed method and experiments interesting and promising. A few questions regarding the contribution, impact of action space, too simple experimental environments, details of the method, limitations, and reasoning of design choices were raised.

The rebuttal addressed all major concerns of the reviewers.

3 of the 4 reviewers have a clear vote for accepting the paper, the last reviewer is leaning slightly towards rejecting the paper with concerns about the paper as a "package". While the paper certainly has its limitations (also pointed out by the authors), I believe it makes an interesting contribution and step forward that is worthwhile presenting to the community.